# Multiple serine transposase dimers assemble the transposon-end synaptic complex during IS607-family transposition

Wenyang Chen[1], Sridhar Mandali[1], Stephen P Hancock[1†], Pramod Kumar[1‡], Michael Collazo[2], Duilio Cascio[2], Reid C Johnson[1,3*]

[1]Department of Biological Chemistry, David Geffen School of Medicine, University of California at Los Angeles, Los Angeles, United States; [2]Department of Energy Institute of Genomics and Proteomics, University of California at Los Angeles, Los Angeles, United States; [3]Molecular Biology Institute, University of California at Los Angeles, Los Angeles, United States

*For correspondence:
rcjohnson@mednet.ucla.edu

Present address: †Department of Chemistry, Towson University, Towson, United States; ‡National Center for Cell Science, Maharashtra, India

Competing interests: The authors declare that no competing interests exist.

**Abstract** IS607-family transposons are unusual because they do not have terminal inverted repeats or generate target site duplications. They encode two protein-coding genes, but only *tnpA* is required for transposition. Our X-ray structures confirm that TnpA is a member of the serine recombinase (SR) family, but the chemically-inactive quaternary structure of the dimer, along with the N-terminal location of the DNA binding domain, are different from other SRs. TnpA dimers from IS1535 cooperatively associate with multiple subterminal repeats, which together with additional nonspecific binding, form a nucleoprotein filament on one transposon end that efficiently captures a second unbound end to generate the paired-end complex (PEC). Formation of the PEC does not require a change in the dimeric structure of the catalytic domain, but remodeling of the C-terminal α-helical region is involved. We posit that the PEC recruits a chemically-active conformer of TnpA to the transposon end to initiate DNA chemistry.
DOI: https://doi.org/10.7554/eLife.39611.001

## Introduction

Although sometimes thought of as DNA parasites, transposable elements (TE) are widely recognized as playing prominent roles in the evolution of genomes (*Biémont, 2010*; *Brunet and Doolittle, 2015*; *Volff, 2006*). TE-derived sequences make up almost half of the human genome, and in some organisms like Maize, make up the vast majority of the genome (*International Human Genome Sequencing Consortium et al., 2001*; *Springer et al., 2009*). TEs can be usurped or 'domesticated' to perform critical functions, such as promoting DNA rearrangements essential for immunity in mammals or in the development of the micronucleus in ciliated protozoa (*Baudry et al., 2009*; *Kapitonov and Jurka, 2005*; *Nowacki et al., 2009*). In bacteria, mobile DNA elements promote horizontal spread of pathogenicity determinants and antibiotic resistance genes (*Frost et al., 2005*; *Hooper et al., 2009*). TEs are also exploited for genome engineering (*Ivics and Izsvák, 2010*; *Woodard and Wilson, 2015*).

Transposases have been reported to be the most frequently occurring functional group of proteins (*Aziz et al., 2010*). Among the four major classes of DNA transposases, the large and diverse DDE/D family that contain an RNase H fold and typically transpose through a cut-and-paste mechanism has been the most intensively studied (*Hickman et al., 2010*; *Yuan and Wessler, 2011*). Recently, the mechanism of transposition by HUH-family elements, which undergo a rolling circle replicative mechanism of DNA transfer, has been elucidated (*He et al., 2015*). The tyrosine- and serine-family of recombinases, which have been extensively studied in the context of site-specific

recombination reactions, also promote DNA transposition. Respectively, these enzymes splice DNA through a sequential pair of single-strand exchanges or through double strand breaks, generating a transient covalent linkage between the cleaved DNA end and a tyrosine or serine on the protein (*Rubio-Cosials et al., 2018*; *Stark, 2014*; *Wood and Gardner, 2015*). In this study, we investigate the mechanism by which the IS*607*-family of serine recombinases transpose DNA. As described below, IS*607*-family TEs have a number of properties that are unusual among TEs, and the serine transposase structure has features unlike other serine recombinases.

Serine recombinases (SRs) have been broadly classified into three subfamilies (*Smith and Thorpe, 2002*). The small SRs (smSR) typically catalyze highly regulated recombination reactions between specific DNA sites that are usually on the same DNA molecule (*Johnson, 2015*; *Rice, 2015*). The serine integrase or large SR (LSR) subfamily typically promote phage integration and excision between specific sites (*Smith, 2015*; *Van Duyne and Rutherford, 2013*), but certain members promote DNA translocation reactions (*Bannam et al., 1995*; *Wang et al., 2006*). SmSRs and LSRs have their DNA binding domains (DBDs) at the C-terminal end of the protein, albeit the LSRs have a more elaborate C-terminal DNA binding and regulatory domain. The SRs found in IS*607*-family transposable elements, however, are distinguished by the location of their DBDs at their N-termini (experimentally confirmed below). This domain architecture is paradoxical because studies on smSRs imply that an N-terminally located DBD would be incompatible with the formation of active tetramers, which is the critical regulatory step of these reactions (*Johnson, 2015*; *Rice, 2015*; *Stark, 2014*).

The founding member of the IS*607* family was first described by Berg and co-workers (*Kersulyte et al., 2000*), who also noted the relationship between the *Helicobacter pylori* IS*607* element and annotated insertion sequence elements like IS*1535* in the *Mycobacteria tuberculosis* genome sequence (*Cole et al., 1998*). IS*607*-family elements have been subsequently found in a wide range of bacterial species, including cyanobacteria, and in archea (*Filée et al., 2007b*; *Kuno et al., 2010*). IS*607*-related sequences have also been found in eukaryotic genomes and viruses, probably primarily through horizontal DNA transfer events, and have been described as the most widely distributed transposon in nature (*Filée et al., 2007a*; *Gilbert and Cordaux, 2013*).

IS*607* elements encode two *orfs*, which often overlap in their coding sequence (*Figure 1A*). OrfA exhibits homology with SRs and is sufficient to mediate transposition of IS*607* in *E. coli* (this paper and *Kersulyte et al., 2000*). The OrfB sequence bears a clear relationship with RuvC and Cas9, and is also present in some IS*200*/IS*605* family members, some eukaryotic transposons, and as stand-alone genes (*Bao and Jurka, 2013*; *Kapitonov et al., 2015*). Surprisingly, the DNA sequences at the termini of individual IS*607* elements are not related, but an inverted repeat sequence, often imperfect, is present near but at different distances from the ends of the element (*Figure 1B*). A common feature of the ends of IS*607* elements is the presence of short directly-repeated motifs, which are positioned at different spacings with respect to each other and to the host DNA junction (*Figure 1B*). An additional unusual feature is that the IS*607* transposition reaction does not create target site duplications (this paper and *Blount and Grogan, 2005*; *Kersulyte et al., 2000*). The absence of target site duplications may make IS*607* elements useful as vehicles for delivering and subsequently removing genes from chromosomes without generating a genetic scar, similar to applications of the TE *piggyBac* (*Woltjen et al., 2009*; *Woodard and Wilson, 2015*).

In this work we investigate the serine transposase from three IS*607*-family elements: IS*607* from *H. pylori* (*Kersulyte et al., 2000*), IS*1535* from *M. tuberculosis*, and ISC*1926* from the hypermophilic archea *Sulfolobus islandicus* (*Blount and Grogan, 2005*). We confirm that OrfA is the only IS*607*-encoded protein required to catalyze transposition in *E. coli*, determine the domain structure of the three transposases, and describe X-ray structures of the OrfA catalytic domains from IS*1535* and ISC*1926*, which exhibit remarkable differences in quaternary structure from other SR-family members. We show that OrfA from IS*1535* efficiently generates paired-end complexes by an unexpected mechanism involving cooperative assembly of multiple proteins, which is both unlike other transposases studied to date and unlike synaptic complex formation by other SR-family members.

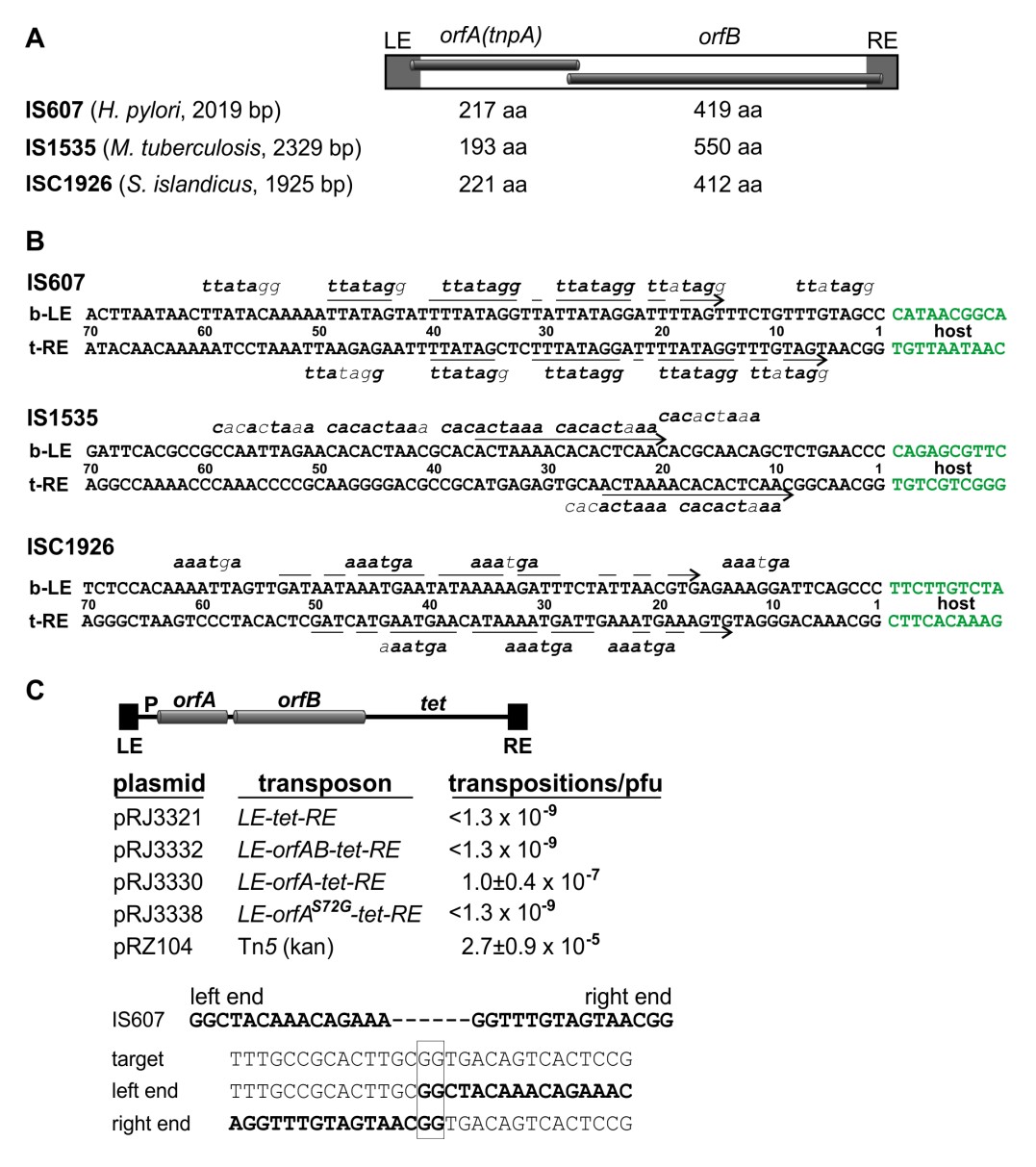

**Figure 1.** IS*607*-family transposons. (A) Overall structure of IS*607*-family transposons with lengths given for the *orfA* and *orfB* coding regions (amino acid residues) of elements discussed in this paper. (B) DNA sequences at the ends of IS*607*-family transposons. The bottom strand of the left end (b–LE) and top strand of the right end (t–RE) are aligned with flanking host DNA sequences in green. Arrows highlight common sequences (inverted repeats) between the ends, and short sequence motifs (bold type are matches) for individual elements are denoted above and below the end sequences (for IS*607* and ISC*1926*, sequence motif lengths can be extended with A or T on either side). The transposon-host borders for each of these elements have been reassigned based on alignments with related elements in their respective genomes and sequence analysis of transposition events (IS*607* and ISC*1926*). The termini contain a GG, and the unoccupied host target sequences also contain a GG at the exchange site (e.g., panel C). (C) Transposition by IS*607* in *E. coli*. Top: reconstructed IS*607* transposons used in the transposition assays. *OrfA* and *orfB,* when present, are transcribed from the *E. coli lac* promoter (P) and contain ribosome binding sites. Middle: transposition frequencies onto phage λ of IS*607* derivatives. Average and standard deviations are given for IS*607orfA* (n = 6) and Tn*5* (n = 3) as a comparative control. Bottom: an example of a λ::IS*607orfA* transposition product. Sequences of the IS*607* ends (bold), the unoccupied target, and the left and right end junctions after insertion of IS*607orfA* are shown. The site of DNA exchange is boxed. Additional insertion site sequences and a compilation are given in *Figure 1—figure supplement 1*.

DOI: https://doi.org/10.7554/eLife.39611.002

The following figure supplement is available for figure 1:

**Figure supplement 1.** Sequence analysis of IS*607* insertion sites.

DOI: https://doi.org/10.7554/eLife.39611.003

## Results

### IS607 transposition in vivo

We first sought to confirm and extend salient features of IS607 transposition originally described by Berg and co-workers (*Kersulyte et al., 2000*). We engineered tetracycline-resistant IS607 derivatives containing the left and right transposon ends and *orfA* or *orfA+orfB* genes (*Figure 1C*). Transposition onto λ was measured after phage induction in a *recA E. coli* λ lysogen, and the resulting λ lysates were used for transduction selecting tetracycline resistance. λ::IS607-*tet* transpositions were obtained for IS607*orfA* at a frequency of $1 \times 10^{-7}$/pfu (*Figure 1C*), but no confirmed transposition events were obtained with IS607*orfAB*. We note that the relative expressions of *orfB* and *orfA* in the IS607 constructs are likely to be different than in the native element; nevertheless, these results indicate that OrfA is sufficient for promoting transposition and that OrfB is inhibitory, as concluded earlier (*Kersulyte et al., 2000*). No transposition events were obtained when OrfA contained a glycine substituted for the predicted active site serine (residue 72), consistent with OrfA catalyzing the transposition reaction through an SR mechanism. The frequency of IS607*orfA-tet* transposition into λ DNA was about 0.4% of that measured for the well-characterized transposon Tn5.

PCR analysis of the λ::IS607*orfA-tet* insertions confirmed the events were simple insertions and sequences of the new transposon-host boundaries showed that all insertions were at a GG dinucleotide target with no duplications of host sequence at the junctions (*Figure 1C* and *Figure 1—figure supplement 1*). A compilation of transposition events promoted by IS607*orfA* (this work) and IS607*orfAB* elements (*Kersulyte et al., 2000*) show that a (G)GG sequence is a preferred target, but no additional sequence relationships among the targets are evident (*Figure 1—figure supplement 1*). A GG dinucleotide at the transposon termini, together with an invariant GG at the insertion target site, is consistent with a DNA exchange reaction over a 2 bp identical sequence that is observed for other SRs.

### IS607-family TnpA domain architecture

The in vivo studies indicate that OrfA, hereafter called TnpA, is the only IS607 protein required for transposition. Purified preparations of recombinant TnpA from IS607, IS1535, and ISC1926 were obtained, and each protein was shown to be active for DNA binding to its cognate transposon ends (below and not shown). To probe domain architectures, each TnpA was subjected to partial proteolysis under native conditions followed by SDS-PAGE and mass spectrometry (*Figure 2A* and *Figure 2—figure supplement 1*). In each case, a trypsin-resistant fragment representing the catalytic domain and helix E region attached to a 3- (TnpA$^{ISC1926}$) to 11- (TnpA$^{ISC1535}$) residue N-terminal segment, which is predicted to be unstructured, was generated. Trypsin also cleaves near the middle of the helix E region of TnpA$^{IS1535}$ and TnpA$^{IS607}$ where available crystal structures show a ~ 4 residue turn separating the N- and C-terminal sections of the helix (see below). Structural models (Phyre2) of the N-terminal domains predict winged-helix motifs that closely match protein-DNA structures present in the PDB (*Figure 2—figure supplement 1*).

### IS607-family TnpA structures

X-ray crystal structures for the C-terminal domains (CTDs) of TnpA from IS1535 (residues 51 – 193) and ISC1926 (residues 65 – 221) were determined and found to contain either one or two dimers in their asymmetric unit, respectively (*Figure 2B,C*, *Figure 2—figure supplements 2A–D*; *Table 1*). Each chain adopts a structure that includes four α-helices sandwiching four β-strands from the beginning of the CTD to the end of β4 (TnpA$^{IS1535}$ residues 51 – 144 and TnpA$^{ISC1926}$ residues 65 – 162), a topology that is identical to that of the catalytic core of smSRs (*Figure 2D,E*). Pairwise structure alignments to the end of β4 between the catalytic domains of TnpA$^{IS1535}$ and TnpA$^{ISC1926}$ and the smSRs γδ resolvase (PDB code 1GDT) and Sin (PDB code 2R0Q) dimers give rms deviations from 1.6 to 3.3 Å, even though pairwise sequence comparisons between the catalytic domains TnpA proteins and smSRs typically exhibit <30% amino acid identity (with short indels).

Although there is considerable structural similarity between the catalytic core domains of the subunits, the quaternary structures of the TnpA and smSRs dimers are radically different. The dimerization interface of the core domains of TnpA is between helices B and D of each subunit (*Figure 2B, C*), which do not share contacts in the smSR dimer structures (e.g., *Figure 2D*). The 961 Å²

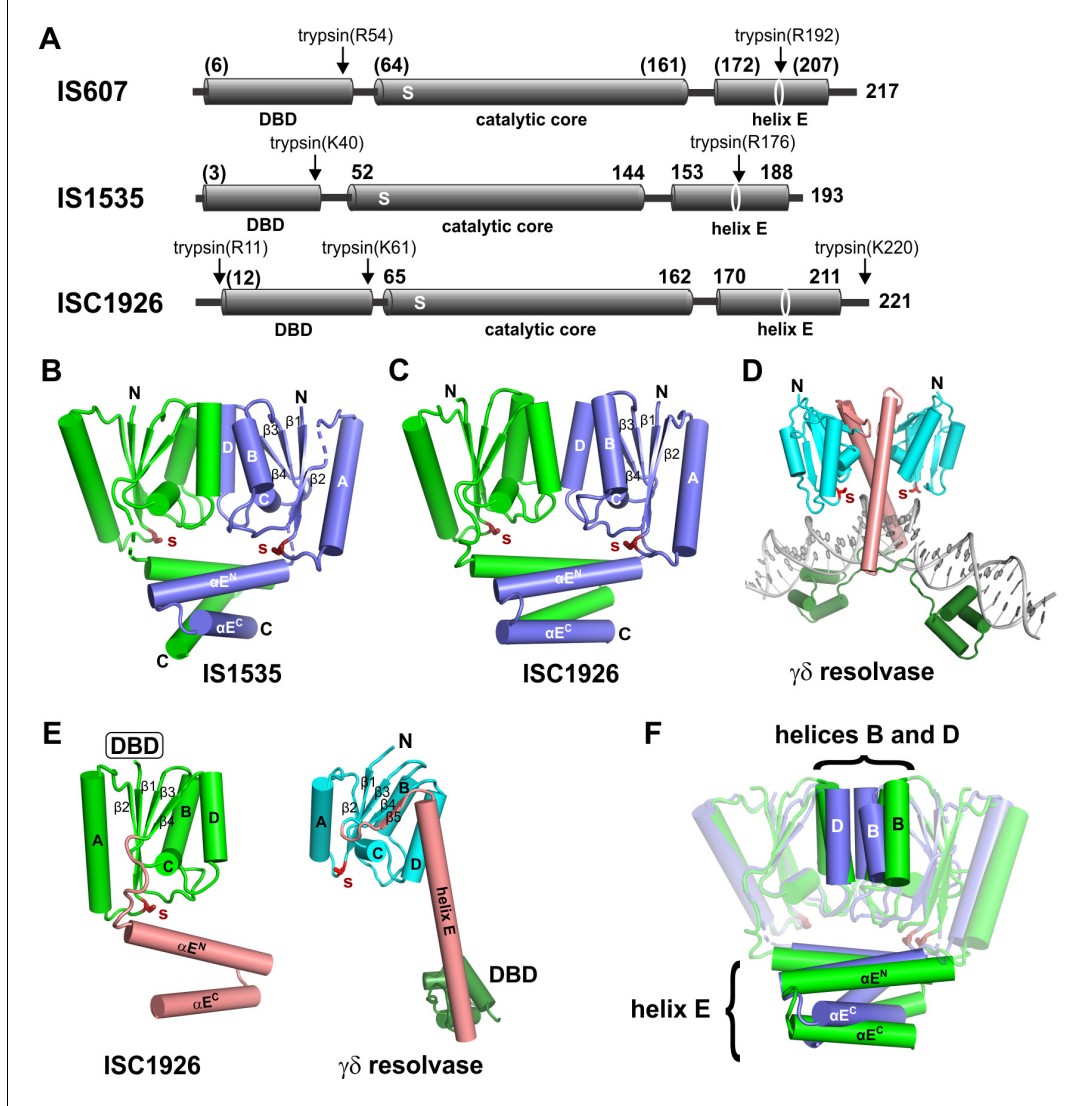

**Figure 2.** Structures of TnpA proteins. (**A**) Domain architecture of TnpA proteins. Domain structures were derived from partial proteolysis/mass spectrometry (*Figure 2—figure supplement 1*), X-ray crystallography for TnpA[IS1535] and TnpA[ISC1926], and Phyre2 models for the N-terminal DBDs (*Figure 2—figure supplement 1*) and the TnpA[IS607] CTD. S denotes the predicted active site serine residue. (**B and C**) X-ray structures of the dimeric CTDs of TnpA[IS1535] and TnpA[ISC1926], respectively. The helix E region folds into a 4-helix bundle that stacks on the catalytic core and occludes the catalytic serines. (**D**) Structure of the smSR γδ resolvase bound to DNA (PDB code: 1GDT). Unlike the TnpA proteins, the dimer interface is over the extended E-helices (salmon), and the DBD (dark green) is at the C-terminus. (**E**) Subunit structures of TnpA-CTD[ISC1926] and γδ resolvase highlighting the common folds of the catalytic cores but different helix E structures. (**F**) TnpA[IS1535] (blue) and TnpA[ISC1926] (green) dimers are aligned over the catalytic domains of subunits A (rmsd = 1.1 Å). Helices B and D at the core dimer interface and the helix E bundles are highlighted to illustrate differences.
DOI: https://doi.org/10.7554/eLife.39611.004

The following figure supplements are available for figure 2:

**Figure supplement 1.** Limited proteolysis and structures of TnpA proteins.
DOI: https://doi.org/10.7554/eLife.39611.005

**Figure supplement 2.** TnpA structures.
DOI: https://doi.org/10.7554/eLife.39611.006

(TnpA[IS1535]) and 924 Å$^2$ (TnpA[ISC1926]) dimer interfaces within the core are relatively flat and hydrophobic, but there are a few polar connections between the subunits. By contrast, smSR subunits are associated in the dimer via their helix E regions through almost exclusively hydrophobic contacts (*Figure 2D*). Although the overall configurations of the TnpA[IS1535] and TnpA[ISC1926] dimers are similar, there are significant differences in the details of the dimer interfaces within the core (*Figure 2F*).

**Table 1.** X-ray diffraction data and refinement statistics.

| Structure PDB code | ISC1926-TnpA 6DGC | IS1535-TnpA – Native 6DGB | IS1535-TnpA – SeMet |
|---|---|---|---|
| **Data collection** | | | |
| Beamline | APS 24 ID-C | APS 24 ID-C | APS 24 ID-C |
| Space group | C1 | $P2_12_12_1$ | $P2_12_12_1$ |
| Unit cell dimensions | | | |
| a, b, c (Å) | 97.1, 212.3, 61.6 | 52.6, 54.2, 104.38 | 52.3, 54.1, 104.5 |
| α, β, γ (°) | 90.0, 126.7, 90.0 | 90.0, 90.0, 90.0 | 90.0, 90.0, 90.0 |
| Wavelength (Å) | 0.9793 | 0.9792 | 0.9792 |
| Resolution range (Å)* | 20 - 2.9 (3.0-2.9) | 48.1 - 2.5 (2.6-2.5) | 52.3 - 2.5 (2.6-2.5) |
| Measured reflections | 71134 | 44467 | 68063 |
| Unique reflections | 19649 | 10275 | 19606 |
| $R_{merge}$[†] | 5.0 (51.1) | 9.9 (64.8) | 7.9 (75.9) |
| $CC_{1/2}$ | 0.99 (0.76) | 0.99 (0.85) | 0.99 (0.82) |
| I/σ | 12.8 (1.3) | 6.5 (1.5) | 10.0 (1.3) |
| Completeness (%) | 88.6 (56.8) | 95.2 (91.4) | 98.8 (95.0) |
| **Refinement** | | | |
| Resolution (Å) | 2.9 | 2.5 | |
| No. of reflections | 15775 | 7951 | |
| $R_{work}$ | 22.0 | 22.8 | |
| $R_{free}$[‡] | 24.6 | 26.1 | |
| RMSD bond length (Å) | 0.01 | 0.01 | |
| RMSD bond angle (°) | 1.15 | 1.17 | |
| No. of atoms | | | |
| Protein | 3996 | 2002 | |
| Water | 0 | 24 | |
| Average B factors | | | |
| Protein | 76.4 | 50.2 | |
| Solvent | | 28.9 | |
| Ramachandran statistics[§] | | | |
| Favored | 97.1 | 95.2 | |
| Allowed | 2.9 | 4.6 | |
| Outliers | 0 | 0.2 | |

*Values in parentheses refer to the highest resolution shell.

[†]$R_{merge} = \Sigma \mid I\text{-}<I> \mid / \Sigma \mid$

[‡]Calculated using 5% (IS1535) and 10% (ISC1929) of the data.

[§]Percentage of residues in Ramachandran plot regions were determined using PROCHECK

DOI: https://doi.org/10.7554/eLife.39611.007

The TnpA[ISC1926] subunits are shifted apart by about 3 Å relative to TnpA[IS1535], the TnpA[ISC1926] D helices are angled by about 15° rather than being parallel, and the TnpA[ISC1926] B helices are one turn longer than in TnpA[IS1535].

The active site serines of each TnpA dimer (TnpA[IS1535] residue 59 and TnpA[ISC1926] residue 74) are separated by 28.6 and 31.5 Å (Cα atoms), respectively (*Figure 2B,C*). This is a much longer distance than would be predicted to catalyze cleavage of scissile phosphates across the minor groove of B-DNA, assuming a 2 bp staggered cleavage (~14 Å separation) that is common to other SRs. An even longer separation between active site serines is present in the catalytically-inactive dimers of γδ resolvase and Sin (*Mouw et al., 2008*; *Yang and Steitz, 1995*).

The helix E regions of the TnpA dimer structures are also completely different from those of other SRs (*Figure 2E*). After β4 in the TnpA dimers, a poorly structured 9 – 10 residue peptide travels along one side of the active site to connect to the helix E region (*Figure 2—figure supplement 2D*). The E helices are interrupted by a four residue β-turn (GRRG in TnpA[IS1535] and GMRS in TnpA[ISC1926]) and fold into an antiparallel structure. The split E helices from each subunit associate into a 4-helix bundle, with the C-terminal segments of TnpA[IS1535] helix E rotated 35° relative to those of TnpA[ISC1926] (*Figure 2F*). The helix E region excludes a total of about 3685 Å$^2$ of solvent accessible surface area in both proteins and would sterically prevent DNA from associating with the active sites (*Figure 2B C F*). The helix E conformation and the separation of active site serines indicate that this dimer conformation cannot be active for DNA chemistry (see also *Boocock and Rice, 2013*).

The structures of the IS1535 and ISC1926 TnpA dimers are very similar to SRs from *Methanocaldococcus jannaschii* (PDB code 3LHK;) and *Sulfolobus solfataricus* (PDB codes 3ILX and 3LHF) (*Figure 2—figure supplement 2E*), which have been discussed previously (*Boocock and Rice, 2013*). Nevertheless, because the quaternary structures of the TnpA-like proteins are so different from other SRs and because these differences have profound functional implications, we tested aspects of the dimeric structure by cysteine crosslinking. Cysteines were substituted at TnpA[IS1535] residues within the catalytic core and helix E region where they would be proximal and oriented appropriately for intersubunit disulfide formation (*Figure 2—figure supplement 2F*). F126C, located just before the start of helix D, and Q138C at the C-terminal end of helix D efficiently formed dimers after oxidation. Within the helix E bundle, L162C generated substantial amounts of covalently-linked dimers and A182C generated a small amount of dimers after oxidation. These solution results substantiate the dimeric structures observed by crystallography.

## Binding of TnpA to the transposon ends

DNA binding by full-length TnpA proteins of IS607, IS1535, and ISC1926 to their respective transposon ends was observed by gel mobility shift assays (EMSAs). Binding by the TnpA[IS1535] to its left end (LE) was the most robust so we focus on IS1535 in the analysis below. As expected, no DNA binding was observed for TnpA[IS1535] missing residues 1 – 50 comprising the N-terminal winged-helix domain.

*Figure 3A* shows complexes formed with increasing amounts of TnpA[IS1535] incubated with radiolabeled DNA probe of the IS1535 LE plus adjacent host DNA and separated by native PAGE. Formation of a slowly migrating complex is accompanied by loss of the free LE probe. We show in *Figure 3F* (lanes 2 – 10) that the slowly migrating complex contains two LE DNA segments (i.e., a pair-end complex, PEC) by incubating TnpA with the radiolabeled 140 bp probe plus excess unlabeled 240 bp LE fragments. This results in formation of a supershifted complex, demonstrating the presence of both the labeled and unlabeled LE DNA fragments. Most of the LE probe associates into PECs with <10 nM TnpA (*Figure 3A,B*). A much lower level of a complex (complex 1) that accumulates with increasing TnpA concentration is also evident, and a small amount of an additional complex (complex 2) is formed at high TnpA concentrations. Appearance of complex 2 is accompanied by a similar decrease of PECs. Formation of PECs is strongly enhanced by Mg$^{2+}$, Ca$^{2+}$, Mn$^{2+}$, or spermidine; in the presence of EDTA, PEC levels severely decrease and complex 1 coordinately increases (*Figure 3—figure supplement 1*). A time course of PEC assembly on left ends by 10 nM TnpA indicates that PECs form relatively slowly, requiring about 30 min to reach maximum levels (*Figure 3C,D*). Neither time course experiments performed at the optimal 37° or at lower temperatures (not shown), where both rates of formation are slower and yields of PECs are decreased, provide evidence that complex 1 is a kinetic intermediate.

Gel mobility shift assays performed on the right transposon end (RE) generate a different profile (*Figure 3E*). Only a small amount of PECs (3% of total probe) are generated, peaking at 8 nM TnpA, whereas complex 1 continues to increase to become the dominant product at high TnpA concentrations. The RE is also inefficient at forming PECs with the LE (*Figure 3F*, lanes 12 – 19). The poor substrate activity of the RE correlates with the presence of only two sequence motifs (*Figure 1B*).

## TnpA[IS1535] binds over a remarkably long DNA segment in LE-PECs

TnpA[IS1535] LE-PEC assembly reactions were subjected to DNase I footprinting. Protections from DNase I cleavage occurred from LE bp 7 (LE 7; LE 11 on the bottom strand) and extend internally to

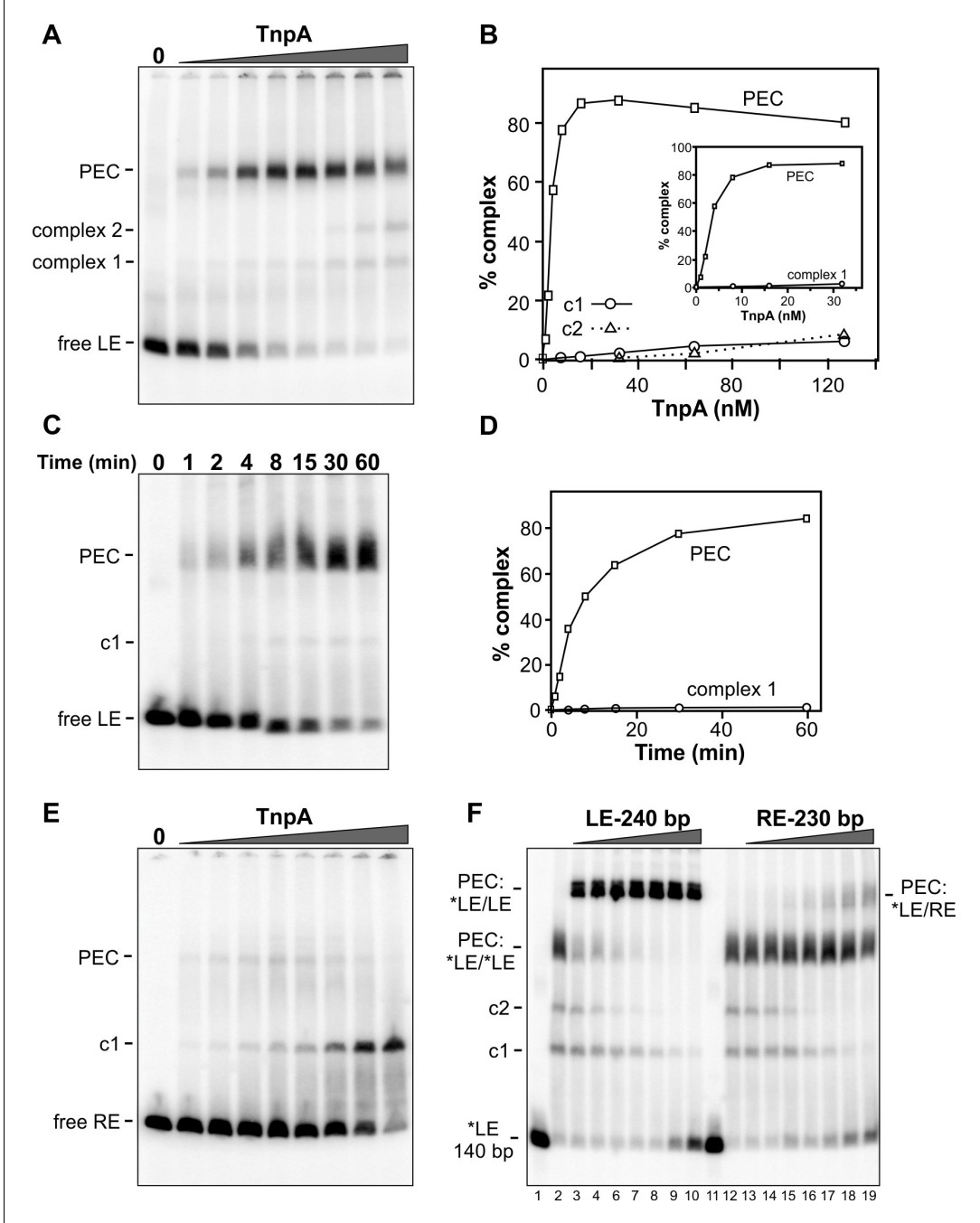

**Figure 3.** Binding of IS*1535* TnpA to transposon ends. (A) Increasing amounts of TnpA (1 to 128 nM in 2-fold increments) were incubated with a 149 bp [32]P-labeled DNA fragment containing the left transposon end and adjacent host sequence. After 1 hr at 37°C, the samples were subjected to native PAGE. The locations of unbound probe (free LE), paired-end complex (PEC), complex 1 (c1) and complex 2 (c2) are denoted. (B) Plot showing relative amounts of the PEC, complex 1, and complex 2 as a function of TnpA concentration. The insert expands the lower TnpA concentration range leading to maximum levels of PECs. (C) Time course of LE-PEC formation. TnpA (8 nM) was incubated with the LE probe at 37°C for increasing times as denoted and applied to a native gel. (D) Plot of the accumulation of LE-PECs and complex 1 as a function of time. (E) TnpA complexes formed on the right end. Reactions were performed as in panel A except that a 139 bp RE DNA probe was used. (F) Formation of hetero-PECs with different lengths LE or RE DNA fragments. In lanes 2 and 12, 100 nM TnpA was incubated with 0.5 nM 149 bp radiolabeled LE probe (*LE). In lanes 3 – 10, increasing amounts of unlabeled 240 bp LE fragments (2 to 128 nM, in 2-fold increments) were included in the reaction. Radiolabeled PECs, but not complex 1 or 2, shift to a slower migrating species in the presence of excess 240 bp LE fragments indicating that these complexes contain both 149 and 240 bp LE DNA molecules. In lanes 13 – 19, increasing amounts of unlabeled 230 bp RE fragments (2 to 128 nM, in 2-fold increments) were included in the reaction with *LE. A small amount of LE + RE PECs form at high RE concentrations. Lanes 1 and 11 are *LE only.

*Figure 3 continued on next page*

*Figure 3 continued*

DOI: https://doi.org/10.7554/eLife.39611.008

The following figure supplement is available for figure 3:

**Figure supplement 1.** IS*1535* PEC formation requires divalent metal ions or spermidine.

DOI: https://doi.org/10.7554/eLife.39611.009

about LE 75 at TnpA concentrations generating PECs (*Figure 4A,E*). DNA sequences over motifs *a-d* show particularly strong protections together with a series of cleavage enhancements that are separated by about 10 bp. The protected region, albeit weaker, continues internally from motif *d* to about LE 75. Clear evidence of TnpA binding over motif *e* is present, but surprisingly weak protections are detectable at nucleotides surrounding the transposon-host junction. Notably, sequences outside of core motifs *a-d* become protected with increasing TnpA concentrations coordinately with the core motifs, implying cooperative binding of TnpA over about 70 bp of the LE concurrent with formation of the PEC (*Figure 4—figure supplement 1*).

Digestion by Exo III, a 3′ to 5′ exonuclease, generates a weak TnpA$^{IS1535}$-dependent stop at LE 8 and a strong stop at LE 18 on the top strand (*Figure 4B,E*). On the bottom strand, Exo III digestion stops occur at LE 78/77 (weak) and LE 68/67 (strong). Increasing Exo III digestion times on LE-PECs suggest that the nuclease can progressively remove 10 bp blocks of TnpA$^{IS1535}$-mediated protection (*Figure 4C*). For example, the weaker stop at LE 8 near the host boundary is nearly lost at long digestion times, and longer digestion times on the bottom strand result in loss of the LE 78/77 stop, increasing amounts of the LE 68/67 stop, and a new product at LE 59/58. Taken together, the Exo III and DNase I footprinting results indicate that strong TnpA binding to the LE occurs between approximately LE 18 and LE 67 with weaker binding extending at least 10 bp in both directions. Both footprinting methods indicate weak, if any, binding over and adjacent to the transposon-host boundary.

## TnpA$^{IS1535}$ binds only over the two motifs on the IS*1535* RE

As described above, TnpA binding to the IS*1535* RE primarily forms a complex I product (*Figure 3E*). Incubation of TnpA-RE reactions with Exo III resulted in digestion stops at RE 7 (top strand) and RE 28 (bottom strand), which flank the two motifs present on this end (*Figure 4D,E*). TnpA was unable to protect the IS*1535* RE from DNase I cleavage, although weak cleavage enhancements were evident at positions within the two motifs that are analogous to the strong enhancements observed in the LE motifs (not shown). The 20 bp Exo III protected region on the RE provides evidence that complex I reflects TnpA binding to two adjacent motifs.

## LE-PEC formation requires IS*1535* motifs a-d plus flanking non-specific DNA sequences

Gel mobility shift assays on probes with progressively truncated endpoints internal to the LE reveal that about 84 bp are required for robust PEC formation (*Figure 5*, top panel, and *Figure 5—figure supplement 1A*). Less efficient PEC assembly is observed with LE segments deleted down to 69 bp, with substrates containing endpoints at LEΔ74 and LEΔ69 generating faster migrating PECs, suggesting fewer molecules of TnpA in the complex. No detectable PECs form with a substrate truncated at LEΔ64. Amounts of complex I generally increase as PEC levels decrease until LEΔ44 where levels of complex I diminish, and LEΔ39, where complex I is not detectable. Addition of non-specific DNA to the LE 39 end restores complex I formation (*Figure 5—figure supplement 1D*).

Resections of host DNA and sequences at the transposon end result in moderately decreasing efficiencies of PEC formation, with LE5Δ, which removes 4 bp of the transposon end, requiring about 10-fold more TnpA than full length substrates (*Figure 5*, middle panel, and *Figure 5—figure supplement 1B*). Low levels of PECs are generated with LE10Δ, LE15Δ, and LE20Δ, which remove DNA up to the beginning of motif *a*, and PECs are not detectable with LE25Δ, which removes part of motif *a*. As observed with the upstream resections, complex 1 levels increase somewhat as PEC assembly becomes less efficient but decrease markedly with the LE25Δ truncation where motif *a* is disrupted.

The upstream and downstream truncation series define the minimal LE DNA segment required for detectable PEC assembly to be between LE 69 and LE 20. These boundaries are consistent with

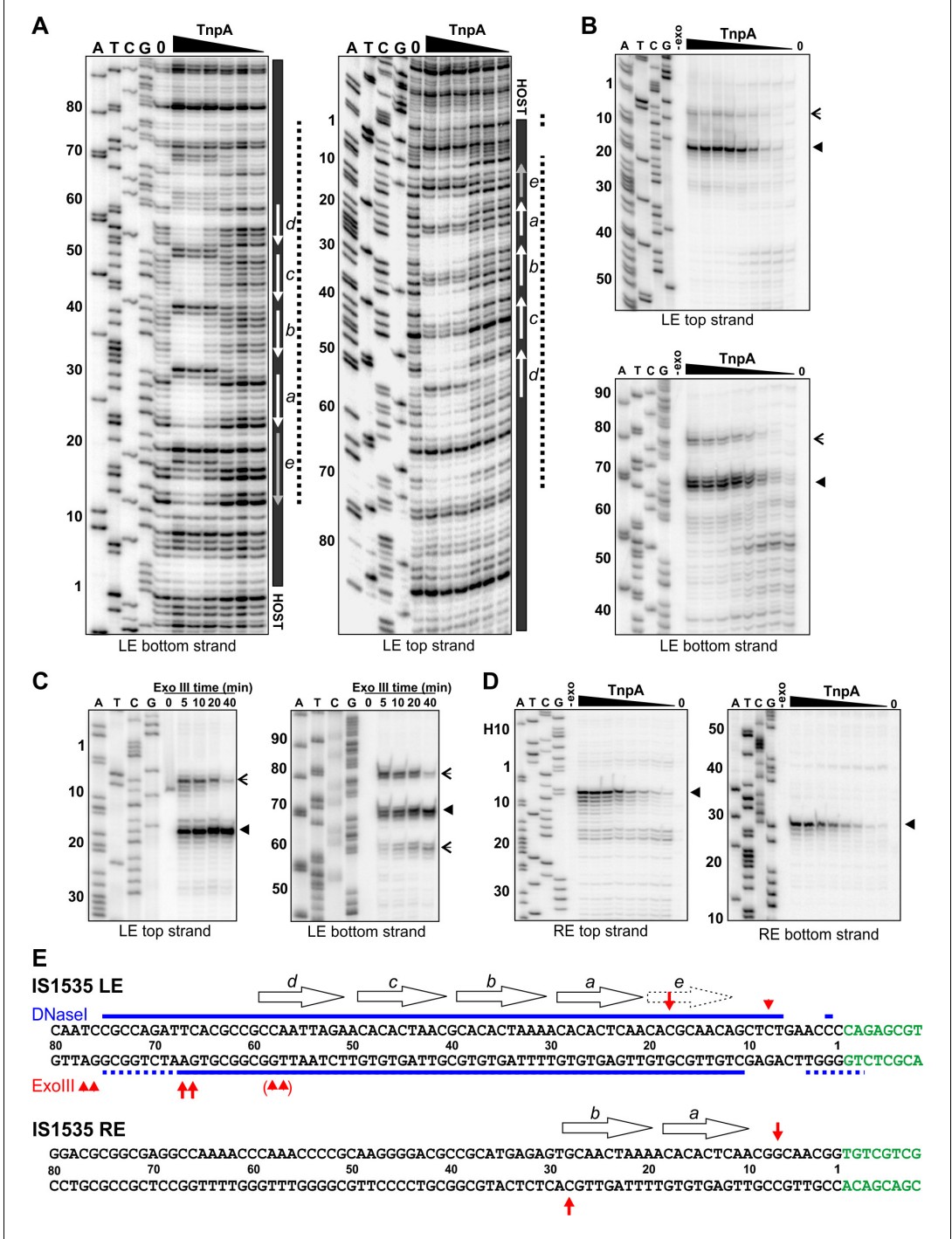

**Figure 4.** Footprint analysis of IS*1535* TnpA binding to the transposon ends. (A) DNase I footprints of TnpA to 5′ end-labeled bottom (left panel) and top (right panel) strands of the LE. TnpA concentrations were from 4 to 128 nM, in 2-fold increasing concentrations, 0 is no TnpA added, and ATCG are dideoxy sequencing lanes primed by the same oligonucleotide used to prepare the footprinting probe. Numbers on the left denote transposon sequence coordinates and are positioned relative to the 0 lane. The black bar on the right marks transposon sequences with arrows showing motif locations. The dashed line denotes regions of significant changes in DNase I cleavage by TnpA. See *Figure 4—figure supplement 1* for EMSAs of binding reactions just prior to DNase I digestion showing relative amounts of PECs. (B) Boundaries of TnpA binding to the LE delineated by Exo III digestion. PEC-assembly reactions, containing from 1 to 128 nM TnpA in 2-fold increasing concentrations, were incubated with Exo III for 30 min. Lane 0 is no TnpA and -exo is no Exo III added. Solid arrowheads indicate major Exo III digestion stops, and open arrowheads denote minor Exo III stops that are TnpA dependent. (C) Time course of Exo III digestion on LE PECs. Preassembled PECs were subjected to Exo III digestion for 0 – 40 min as labeled. (D) Exo III digestion stops on the RE. Reactions were the same as in panel B except that 5′ end-labeled DNA probes representing the RE DNA

*Figure 4 continued on next page*

*Figure 4 continued*
strands were used. (**E**) Summary of DNase I and Exo III footprinting data on the LE and RE sequences. Changes in DNase I reactivity by TnpA are denoted with blue lines; dashed lines indicates weak protection. Red arrows denote Exo III digestion stops; shorter arrows signify minor stops and arrows in parentheses are stops appearing after long digestion times. IS*1535* end sequence motifs (open arrows) are positioned above the sequence.
DOI: https://doi.org/10.7554/eLife.39611.010
The following figure supplement is available for figure 4:

**Figure supplement 1.** EMSA of TnpA[IS1535] binding in DNase I footprint reactions.
DOI: https://doi.org/10.7554/eLife.39611.011

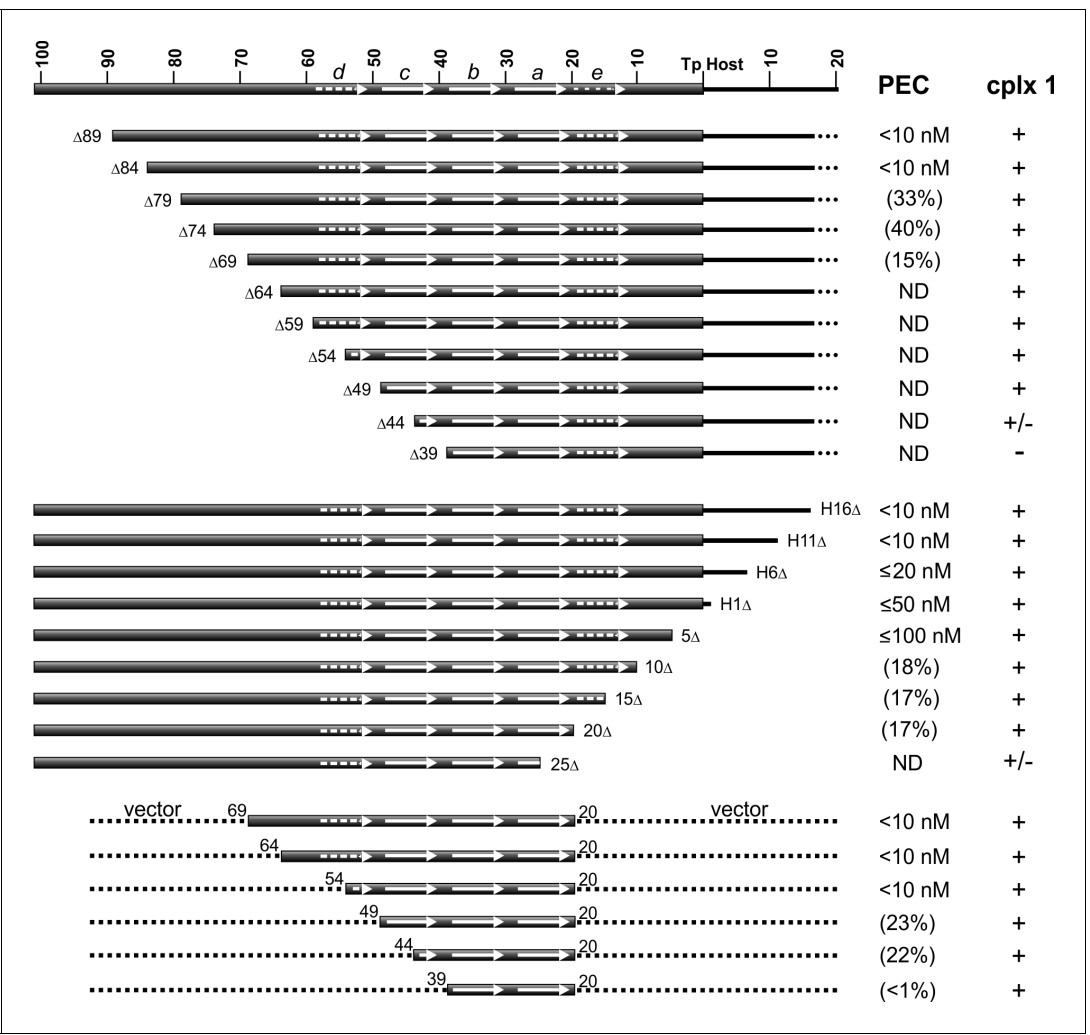

**Figure 5.** DNA sequence requirements for IS*1535* LE-PEC assembly. Top series are LE truncations beginning internal to the transposon. Middle series are truncations beginning within host DNA (H) flanking the transposon. Bottom series are LE sequences from transposon nt 20 to various internal endpoints embedded in vector DNA. PEC assembly was averaged from at least three different experiments for each probe. The concentrations of TnpA[IS1535] required for 50% conversion of the probe to PECs are listed; if <50% of the probe was converted to PECs, the maximum yield of PECs obtained over the TnpA titration series (up to 128 nM TnpA) is given in parentheses. ND indicates PECs are not detected in the EMSAs. The presence of complex 1 is denoted by +, absence by -, and barely detectable levels by +/-. See *Figure 5—figure supplement 1* for supporting data.
DOI: https://doi.org/10.7554/eLife.39611.012
The following figure supplement is available for figure 5:

**Figure supplement 1.** Gel mobility shift assays on IS*1535* LE resections.
DOI: https://doi.org/10.7554/eLife.39611.013

the major Exo III protected borders between LE 67 and LE 18. Appending vector DNA onto the LE20Δ junction (LE20v) fully restores efficient PEC assembly (*Figure 5—figure supplement 1D*). However, appending vector DNA onto the LE25Δ junction did not enable PEC formation or increase levels of complex 1. Although LEΔ64 is inactive for PEC formation (*Figure 5*, top panel, *Figure 5— figure supplement 1A*), appending vector DNA onto the LEΔ64 end (in the context of LE20v) fully restores PEC assembly (*Figure 5*, bottom panel, and *Figure 5—figure supplement 1C*). PEC formation remains efficient on a probe containing transposon sequences down to LE 54 when fused to vector DNA; LE(v54-20v) contains part of motif *d* through motif *a*. Removal of transposon sequences into motif *c* (v49-20v and v44-20v), however, markedly decreases PEC formation, and a substrate containing only LE transposon sequence comprising motifs *a* and *b* (v39-20v) only forms barely detectable levels of PECs but generates complex 1 (*Figure 5*, bottom panel).

Taken together, these results demonstrate that transposon sequences contained in motifs *a-d* (LE 59 to LE 20) encompass the minimal IS*1535* DNA required for efficient PEC assembly. However, PEC formation requires at least an additional 10 bp of non-specific DNA upstream of motif *d* (an additional 25 bp for robust formation), and about 30 bp of nonspecific DNA downstream of motif *a* for fully efficient PEC formation.

## IS*1535* LE core motif sequences nucleate formation of the PEC

LE(v54-20v) efficiently forms PECs even though it contains only 35 bp of transposon sequence corresponding to motif *a* through part of motif *d* (*Figure 5*, bottom panel, and *Figure 5—figure supplement 1C*). TnpA[IS1535] strongly protects sequences from DNase I cleavage on LE(v54-20v) PECs over the core motifs *a-d*, and protections extend into vector sequences on either side of the core motifs at least as far as observed for the native LE (*Figure 6,C*). The major Exo III stops on LE(v54-20v) are at the boundaries of motifs *a-d* (*Figure 6B,C*).

The profile on LE(v54-20v) suggest a mechanism by which TnpA[IS1535] binds cooperatively and with high affinity to the four core motifs *a-d*, even with only half of the native motif *d* sequence present. Additional molecules of TnpA then spread in either direction from the core 'nucleation' segment in a sequence-independent manner. When the motif *d* sequence is completely absent, as in LEΔ49 (*Figure 5*, bottom panel, and *Figure 5—figure supplement 1C*), PEC formation is inefficient, and no PECs are formed when motif *a* is partially removed (LE25v, *Figure 5—figure supplement 1C*).

## The helix E region, but not the TnpA catalytic core domain, is remodeled during PEC assembly

As discussed above, the quaternary structure of TnpA solution dimers are very different from other serine recombinases and are predicted to be in an inactive conformation for DNA chemistry. We asked whether conformational changes were required for cooperative DNA binding and formation of the PEC. Single-cysteine derivatives of TnpA[IS1535] were oxidized to form disulfide-linked dimers and evaluated for their ability to assemble PECs. Cys126 near the N-terminal end of helix D and Cys138 at the C-terminal end of helix D were efficiently oxidized into covalent dimers that lock the two dimeric subunits together within the catalytic domain core (*Figure 7A,B*). Both of these disulfide-linked mutant dimers efficiently formed PECs (*Figure 7C,D*). We conclude that a rearrangement of subunits within the catalytic core of the dimer is not required for PEC assembly.

Cys162, within the helix E region, generated only about 40% disulfide-linked dimers upon oxidation (*Figure 7A,B*). Nevertheless, oxidized Cys162 was completely defective in PEC formation, whereas the reduced cysteine mutant was active (*Figure 7E*). In an additional experiment, we removed the entire helix E region. TnpAΔ(147-193), which contains residues up to the end of β4, was defective for PEC formation (*Figure 7F*). However, at very high protein concentrations a small amount of product migrating as a PEC is observed. Incubation of TnpAΔ(147-193) with the RE fails to generate PECs, as expected, but products migrating as complex 1 are formed (*Figure 7—figure supplement 1*), indicating the truncated protein remains active for forming this species. The properties of the helix E deletion mutant provide evidence that helix E performs an important function in cooperative binding to form PECs but probably not directly in synaptic interactions because a small amount of LE-LE PEC can form. The inability of the covalently-crosslinked E helices to assemble PECs suggests that a conformational rearrangement of the helix E region of the dimer is required, perhaps to enable helix E interactions between adjacent dimers bound to the transposon ends.

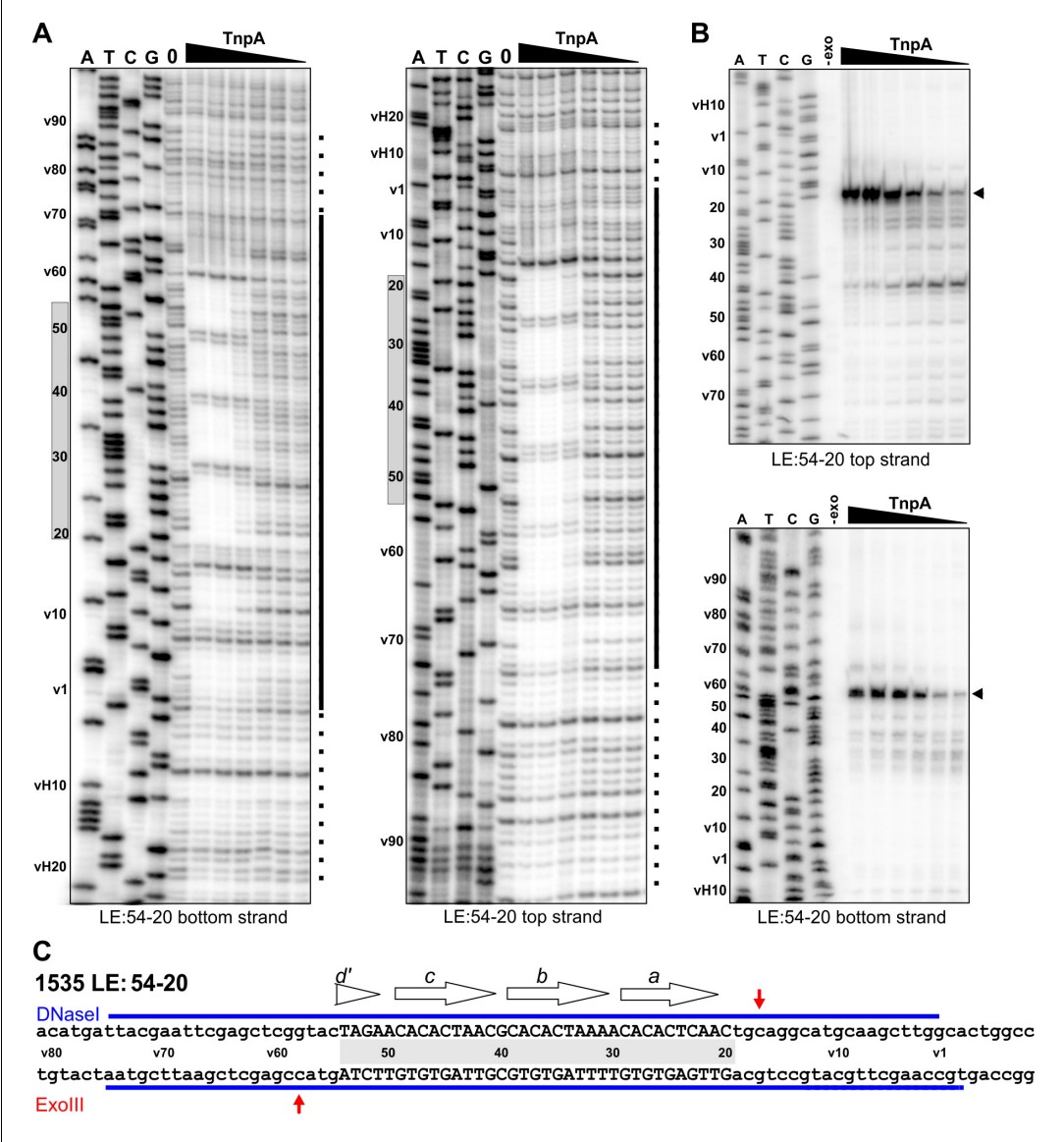

**Figure 6.** Footprint analysis of IS*1535* deletion substrate LE(v54-20v) containing the minimal transposon sequences required for efficient PEC assembly. (A) DNase I footprints of PEC assembly reactions on 5′-³²P-labeled bottom and top strands of LE(v54-20v). TnpA concentrations were from 4 to 128 nM in 2-fold increasing amounts. Shaded rectangles on the left of the gels denote the positions of transposon sequences; coordinates labeled with v are vector sequences with vH being the equivalent locations of host DNA. The bars on the right of the gels denote regions of significant changes in DNase I reactivity by TnpA with dashes indicating weakly protected regions. (B) Exonuclease III delineated boundaries of TnpA binding. TnpA concentrations are the same as in panel A. (C) Summary of DNase I (strongly protected regions, blue) and Exo III digestion boundaries on the LE(v54-20v) sequence. Small letters denote vector sequence.

DOI: https://doi.org/10.7554/eLife.39611.014

## Discussion

The IS*607* family of DNA transposable elements exhibits many features that are not typically found in other transposable elements. The ends of IS*607* elements are not bordered by terminal inverted repeated sequences, and there are no duplications of target sequence at the transposon-host DNA borders; however, multiple short sequence motifs internal to the ends are present (this paper, *Blount and Grogan, 2005*; *Kersulyte et al., 2000*). Most IS*607*-family members terminate in GG, and in the cases of IS*607* and ISC*1926*, insert into a GG target sequence. IS*607*-family elements encode two *orfs* whose coding sequences encompass nearly all of the DNA between the transposon

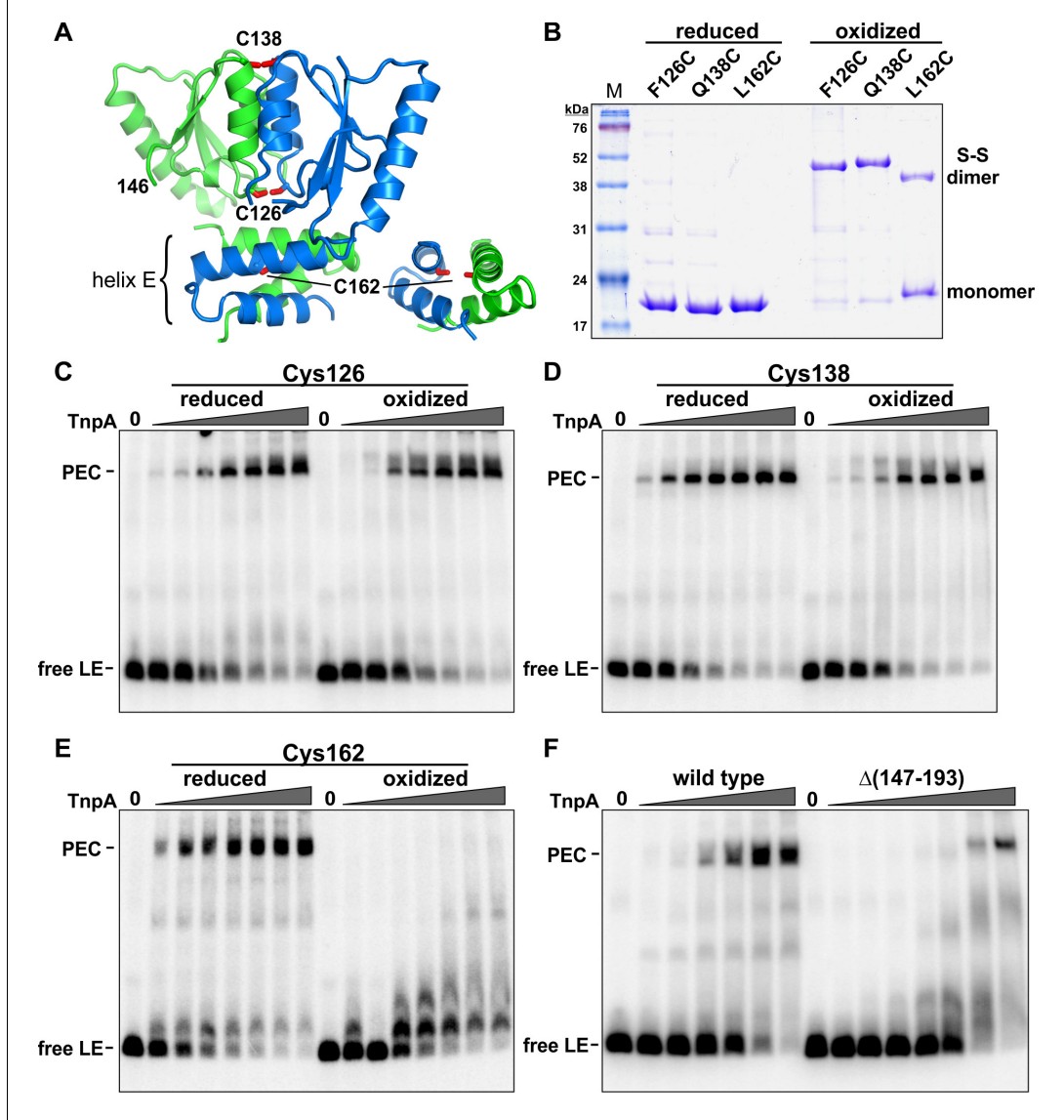

**Figure 7.** Activities of crosslinked TnpA[IS1535] dimers. (A) TnpA[IS1535] dimer structure highlighting residues 126, 138, and 162, which are modeled as cysteines in rotamers compatible for disulfide formation. The helix E region on the right is rotated clockwise in the Y plane about 90° to better visualize Cys162. (B) Non-reducing SDS-PAGE of reduced and oxidized preparations of TnpA[IS1535] mutants containing single cysteine residues. The three native cysteines were replaced with serines in these mutants. (C–E) EMSAs of PEC assembly by reduced and oxidized preparations of Cys126, Cys138, and Cys162 mutants, respectively. The LE probe was incubated with 1 to 64 nM TnpA mutant in 2-fold increasing concentrations. (F) PEC assembly by wild-type TnpA and a deletion mutant missing the helix E region (residues 147 – 193). TnpA concentrations are the same as in panels C-E except that a reaction with 128 nM TnpAΔ(147-193) was included. The location of residue 146 at the C-terminus of this mutant is shown in panel A.

DOI: https://doi.org/10.7554/eLife.39611.015

The following figure supplement is available for figure 7:

**Figure supplement 1.** Formation of IS1535 RE complex 1 with TnpAΔ(helix E).

DOI: https://doi.org/10.7554/eLife.39611.016

ends. For IS607, *orfA/tnpA* is necessary and sufficient for transposition in *E. coli* (this paper and *Kersulyte et al., 2000*). TnpA binds specifically to the transposon ends and is a member of the SR family of DNA exchange enzymes, which are most often associated with site-specific recombination reactions. Residues implicated in catalysis by SRs are well conserved within IS607-family TnpA proteins, and we demonstrate here that the presumed active site serine is required for transposition. Strikingly, however, the dimeric structure is radically different from other SRs. OrfB, whose presence

appears to negatively impact IS607 transposition rates in *E. coli* (this paper and *Kersulyte et al., 2000*), may function as a negative regulator, or perhaps in an ancillary role such as DNA repair, in particular hosts (*Kapitonov et al., 2015*; *Kersulyte et al., 2000*). OrfB-like genes are often associated with IS605/608-family transposons, and OrfB (TnpB) from ISDra2 has also been reported to function as a negative regulator (*Pasternak et al., 2013*).

## Assembly and architecture of the paired-end complex

The first major step in a transposition reaction is formation of a paired-end (synaptic) complex leading to a chemically-active transpososome (*Hickman and Dyda, 2015*). We show that TnpA$^{IS1535}$ binds in a robust and highly cooperative manner to multiple binding sites within the LE of the element and can efficiently recruit a second LE to generate a paired-end complex. Binding nucleates over four 9 bp directly-repeated motifs that are positioned in a helically-phased manner from about 20 to 60 bp from the IS1535 LE terminus (*Figure 1B*). Transposon sequences beginning at motif *a* (LE bp 21) through the conserved half of motif *d* (LE bp 54) are essential for efficient paired-end complex formation. However, additional non-specific DNA sequences extending to about 84 bp from the left end are required for efficient PEC assembly. Likewise, additional DNA extending from motif *a* to shortly beyond the transposon-host junction improves the efficiency of PEC assembly. Although this region contains motif *e*, which is spaced one bp closer to motif *a* than the spacing between motifs *a-d*, the presence of the motif *e* sequence has little discernable effect on PEC assembly. We find it surprising that the sequence identity of the terminal 19 bp of the LE does not significantly influence PEC formation.

Footprinting data on LE-TnpA$^{IS1535}$ PEC complexes are consistent with the LE resections. TnpA$^{IS1535}$ strongly binds over motifs *a-d*, but the overall region of binding extends from before motif *e* to about 75 bp from the LE terminus. Significantly, only very weak binding is evident at sequences surrounding the transposon-host junction where DNA chemistry must occur. PECs formed with LE substrates containing nonspecific sequences downstream of motif *a* actually exhibited greater protections from DNase I cleavage over the region that would be positioned at the transposon border, possibly implying that the native sequence near the LE terminus may be suboptimal for TnpA dimer binding. The boundaries of TnpA binding within LE PECs revealed by Exo III digestion support a model whereby multiple TnpA proteins coat long segments of the left ends. Initial Exo III stops indicate TnpA binding from 8 to 78 bp from the left end terminus. Profiles obtained upon increasing Exo III digestion are consistent with the exonuclease removing TnpA molecules bound to units of about 10 bp, revealing borders of the TnpA nucleoprotein filament from 8 and 18 (major) bp from the host junction extending to 78, 68 (major) and 59 bp within the element.

In contrast to the LE, the IS1535 RE is a poor substrate for TnpA binding. Only a small amount of RE-RE PECs or RE-LE PECs are detectable, although a complex 1 species is formed at high protein concentrations. Exo III footprinting shows that the RE complex 1 contains TnpA bound only to the two sequence motifs that are present between bp 10 and 28 from the RE-host junction. The significance of the differences in the IS1535 ends on its transposition reaction remains to be determined. However, the distribution of sequence motifs, along with our preliminary end binding experiments with TnpA from IS607 or ISC1926, suggests that this disparity is not present in these elements and thus may not be a general feature of IS607-family transposons.

The structures of the IS607-family TnpA catalytic domain dimers pose a number of questions with respect to how DNA binding and catalysis occur, especially in the light of the radically different oligomeric conformations of other SR family members. Whereas the catalytic domains of the smSRs oligomerize via interactions between their helix E regions, TnpA catalytic domains dimerize over their B and D helices. The helix E regions of TnpA dimers are split and folded into a physically separate four helix bundle that is attached to the catalytic domains by a flexible polypeptide linker. The helix E bundle would sterically exclude DNA from associating with the active site. Therefore, minimally, a reconfiguration of the helix E region would be required to enable DNA catalysis. In addition, the active site is located on the opposite side of the subunit from its DNA binding domain (see models below), raising the possibility that cleavage of the synapsed transposon end and/or target DNA may be in trans with respect to the DNA to which the N-terminal DBD is bound (*Boocock and Rice, 2013*), a recurring feature of transpososome structures (*Hickman and Dyda, 2015*). By contrast, smSRs cleave the half site to which they are bound (*Boocock et al., 1995*; *Li et al., 2005*). An additional paradoxical feature with respect to catalysis is that the active site serines in the TnpA dimer

structures are separated by >25 Å, much too far to cleave on either side of the GG dinucleotide at the transposon ends and host target in a manner consistent with other SRs. These and other comparative features with smSRs make it likely that a large conformational change in the oligomeric structure of TnpA precedes DNA cleavage and exchange.

Nevertheless, we show here that the quaternary structure of the catalytic domain is active for assembling PECs, as evidenced by the robust formation of PECs by IS1535 TnpA dimers covalently crosslinked over the core subunit interface. However, the inability of dimers with covalently-linked E helices to cooperatively bind the LE and form PECs provides strong evidence that the helix E region does undergo conformational rearrangement during PEC assembly. An attractive model is that the helix E regions from adjacent dimers remodel to interact with each other during the cooperative loading of proteins along the transposon end. IS1535 TnpA proteins deleted for the entire helix E region appear competent to form PECs at very high protein concentrations, supporting a role for a remodeled helix E in promoting cooperative binding between dimers bound laterally along the transposon ends. The finding that a small amount of PECs appear to still form when the entire helix E region is deleted suggests that helix E is not directly required for synaptic interactions.

Proteolysis experiments and structure modeling suggests that winged-helix DNA binding domains are linked to the N-terminal end of the catalytic domain by peptide chains ranging in length from just three residues (ISC1926) to about 10 residues (IS1535). Structural models of DNA-bound TnpA$^{ISC1926}$ dimers, where there is predicted to be less conformational freedom between the DBD and catalytic domains, are shown in *Figure 8*. In panel A, the recognition α-helices of the two DBDs are inserted into the major groove of a DNA model of the IS1535 LE segment containing motifs *a* and *b* at positions consistent with protections from dimethyl sulfate reactivity at guanines by TnpA$^{IS1535}$ (*Figure 8—figure supplement 1*). The N-termini of the catalytic domains of the TnpA$^{IS1535}$ and TnpA$^{ISC1926}$ dimers are separated by a distance that is close to the pitch of B DNA. Thus, the two DBDs can readily fit into adjacent major groove segments on the same helical face even with the short three residue linker that is present in TnpA$^{ISC1926}$.

An alternative arrangement is shown in *Figure 8B* where only one subunit of the dimer binds to a motif on an individual transposon end, leaving the other subunit free to bind a second DNA. In this model, additional dimers would bind in a similar manner to adjacent motifs (*Figure 8C*) with binding stabilized by remodeling of the helix E regions to generate intermolecular contacts between dimers (e.g., *Figure 8—figure supplement 2*). This dimer binding configuration accounts for the cooperative assembly of TnpA units covering about 10 bp each, as evidenced in the Exo III footprints. Most importantly, it accounts for the near simultaneous recruitment of both DNA ends into a PEC; a TnpA dimer array on a single transposon end will have an array of appropriately spaced free DBDs (*Figure 8D*) ready to capture a second unbound transposon end with high affinity (*Figure 8E*). The parallel arrangement of ends in the PEC is consistent with PEC assembly by a substrate containing two inverted copies of IS1535 LEs separated by only 80 bp (*Figure 8—figure supplement 3*).

We suggest that complex 1, which is formed in vitro at high TnpA concentrations and does not appear to be a precursor to the PEC (*Figure 3A–D*), has a TnpA dimer bound in the conformation depicted in *Figure 8A*. In support of this, a mixture of IS1535 TnpA and TnpA-MBP dimers to a LE deletion substrate that only forms complex I generates only DNA-bound products representing the two homodimers (*Figure 8—figure supplement 4*). If complex 1 consisted of two dimers bound as in *Figure 8B,* a heterotetrameric species that would migrate between the complexes formed by the separate dimer reactions would also be expected.

Complex 2, which also contains only one transposon end, is observed only at very high concentrations of TnpA relative to transposon ends (*Figure 3A,B*). The presence of complex 2 is correlated with a decrease of PECs as the TnpA to transposon end ratio increases (*Figure 3A,B,F*). We propose that this complex has dimers bound along individual LEs as in *Figure 8C,D*, but unbound transposon ends are unavailable to generate a PEC, consistent with a model whereby TnpA bound to one end captures a second unbound end. Alternatively, complex 2 could have two dimers bound in the conformation in *Figure 8A* to motifs *a-d*.

## Comparison with other transpososome structures

Many transposases, for example Tn5, function as dimers where each subunit associates with the short terminal inverted repeats of both DNA ends within the assembled transpososome (*Davies et al., 2000*). However, some transposases utilize multiple binding sites within longer end

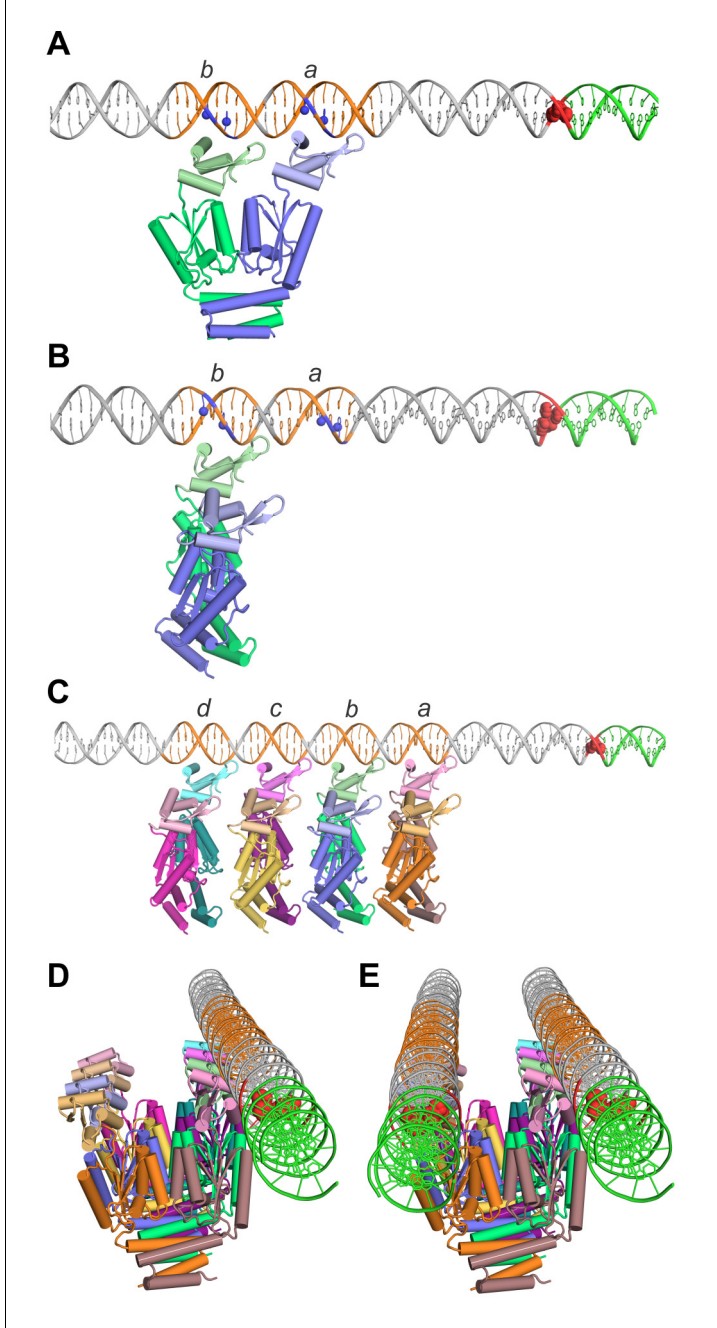

**Figure 8.** Models of TnpA binding to the IS*1535* LE and the PEC. (**A**) Model of a TnpA dimer in a configuration where the two DBDs are binding to LE motifs *a* and *b* (orange DNA) in a manner consistent with DMS protection data (*Figure 8—figure supplement 1*) and where the wing is over the A/T-rich minor groove. Guanine N7 atoms protected from DMS reactivity by bound TnpA are highlighted as blue spheres. The tandem G/C base pairs at the LE terminus are red and the host DNA is green. The TnpA dimer model is derived from the Phyre2 model of the TnpA[ISC1926] NTD (*Figure 2—figure supplement 1C*) linked by three residues to the TnpA[ISC1926] CTD X-ray structure. We posit this conformation on DNA represents complex 1 (see *Figure 8—figure supplement 4*). (**B**) TnpA dimer configuration where only one DBD is associated with a single end. The dimer is rotated orthogonally about the DBD-CTD linker in relation to the dimer in panel A. (**C**) Four TnpA dimers are bound as in panel B to motifs *a-d* on one LE. The helix E regions are proposed to engage in helix-swapped interactions between adjacent dimers (e.g., *Figure 8—figure supplement 2*) to promote cooperative binding. This structure, with additional dimers bound laterally along the LE, may reflect complex 2. (**D**) Model in panel (**C**) rotated to show the DNA in an end-on view, highlighting the set of unbound DBDs. (**E**) Model of the PEC with a second LE associated. *Figure 8 continued on next page*

*Figure 8 continued*

Although represented as parallel straight DNAs, the two transposon ends may be in a more interwrapped structure. TnpA promoters in a different, chemically-active, conformation are proposed to be recruited to the end of the filament at the transposon-host junction.

DOI: https://doi.org/10.7554/eLife.39611.017

The following figure supplements are available for figure 8:

**Figure supplement 1.** Dimethyl sulfate (DMS) protection assay on the IS*1535* LE to identify major groove (guanine N7) contacts.

DOI: https://doi.org/10.7554/eLife.39611.018

**Figure supplement 2.** Illustration of a potential helix-swap mechanism by which the helix E region could remodel to promote cooperative binding onto the LE.

DOI: https://doi.org/10.7554/eLife.39611.019

**Figure supplement 3.** Parallel arrangement of DNA strands within the PEC.

DOI: https://doi.org/10.7554/eLife.39611.020

**Figure supplement 4.** Complex I is a dimer bound to the IS*1535* LE.

DOI: https://doi.org/10.7554/eLife.39611.021

---

segments, and in some cases, contain distinct DNA binding domains. For example, each subunit of the *Mos1* Mariner-family transposase dimer binds to one transposon terminus and additional separate DNA binding domains associate with two subterminal binding sites on the other transposon end within the active complex (*Richardson et al., 2009*). Assembly of the phage Mu transpososome, which contains four copies of the Mu A protein, is more complex as it involves interactions with a remote enhancer-like element by a distinct DNA binding domain (*Harshey, 2014*). *hAT*-family transposons often have many subterminal repeats at variable spacings and orientations, which in some elements, can be located hundreds of base pairs from the transposon termini (*Atkinson, 2015*). The transposase from the *hAT* element *Hermes* is a preassembled donut-shaped octamer that is proposed to bind to four subterminal repeats on each end along with the transposon termini using distinct DNA binding regions (*Hickman et al., 2014*). The presence of subterminal repeats on *hAT*-family elements bears some similarity to the sequence motifs of IS*607*-family transposon ends, but the manner of transposase binding and synapsis proposed here for IS*1535* is different.

## Current understanding of the IS607-family transposition reaction

We propose the following pathway for formation of the paired-end complex that is expected to be a critical early intermediate in the IS*607*-family transposition reaction. Multiple TnpA dimers are initially targeted to DNA motifs that can be located over 60 bp from the transposon end (*Figure 1B*). In the case of the IS*1535* LE, four dimers cooperatively bind over four helically-phased motifs beginning 20 bp from the end (*Figure 8C*), and the nucleoprotein filament continues to spread in a largely sequence-neutral manner to cover at least 70 bp. The active sites are positioned well away from the DNA in the filament, avoiding any spurious cleavage. The single-end complex then captures an unbound end to generate a stable PEC (*Figure 8D,E*). Because formation of the PEC is relatively slow (*Figure 3C,D*), we imagine that initial binding to a IS*1535* LE limits the rate of PEC formation, but once the single-end filament is formed, an unbound end is rapidly captured. Although we illustrate this complex as two parallel transposon ends, a more interwrapped structure may form. No quaternary change in the dimer structure over the catalytic core domain is required for PEC assembly, but our evidence indicates that the helix E region remodels to facilitate cooperative assembly of the nucleoprotein filament. Most all IS*607*-family transposon ends have multiple motifs (the IS*1535* RE is an exception with only two), but they are not always spaced by increments of 10 – 11 bp (*Figure 1B*). The flexible peptide linkers between the DBD and catalytic domains and between the catalytic domains and helix E regions may enable similar nucleoprotein filaments to assemble, even if some of the recognition motifs are not in helical phase.

Whereas an 80 bp segment within the IS*1535* LE beginning near the transposon terminus is required for maximally efficient PEC assembly, only very weak TnpA binding is observed over the transposon-host DNA junction where TnpA-mediated chemistry on DNA must occur. For the reasons discussed above, and because the active sites in the solution dimer are not positioned appropriately with respect to the terminal GG cleavage sites, an alternate conformation of TnpA is almost certainly

required for DNA catalysis. Recruitment of TnpA in a catalytically-active conformer may be a key regulatory step and could require co-translational synthesis or folding localized to a preformed PEC (*Duval-Valentin and Chandler, 2011*). This could explain the requirement in vivo for the IS*607 tnpA* gene to be located close to the transposon ends for transposition to occur (*Kersulyte et al., 2000*) (W.C. and R.C.J., unpublished). Thus, correct assembly of the PEC, with the two transposon ends in correct register, may be a prerequisite or checkpoint for recruiting catalytically-active subunits to bind to the junction, or alternatively, for allosterically activating weakly bound subunits over the junction.

In addition to the structure of the recombination complex, the steps of DNA exchange by serine transposases may also be quite different from other SRs. A subunit rotation mechanism for strand exchange on the complex depicted in *Figure 8E* would lead to an intramolecular inversion, not transposition. Instead, we suggest that the element may excise from the donor site and then insert into a target locus using serine chemistry without strand transfer coupled to subunit rotation. It is also possible that capture of the target locus could be a prerequisite for DNA cleavage. Because both transposon ends need to recombine into a single GG target, it seems likely that the strand transfer reactions must occur sequentially. As none of these DNA cleavage–transfer steps would necessarily require a subunit rotation reaction, the structure of chemically-active TnpA oligomers may be very different from other SRs that have been trapped in tetrameric structures competent for DNA exchange by subunit rotation.

# Materials and methods

### Key resources table

| Reagent type (species) or resource | Designation | Source or reference | Identifiers | Additional information |
|---|---|---|---|---|
| Gene (*Mycobacterium tuberculosis*) | IS*1535 orfA/tnpA* | H37Rv genome DNA | Gene ID: RV0921 | |
| Gene (*Helicobacter pylori*) | IS*607 orfA/tnpA* | synthetic gene | NCBI protein ID: AAF05600.1 | |
| Gene (*Helicobacter pylori*) | IS*607 orfB* | synthetic gene | NCBI protein ID: WP_001274345.1 | |
| Gene (*Sulfolobus islandicus*) | ISC*1926 orfA/tnpA* | *S. islandicus* genome DNA, PMID: 15612937 | NCBI protein ID: AAV87873.1 | *S. islandicus pyrE::ISC1926* Dennis Grogan, University of Cincinnati |
| Strain, strain background (*E. coli*) | RJ1224 | Laboratory collection | *recA56, srl, Δ(pro-lac), ara, rpsL, λbbnin [λ cI857,b515, b519, nin5, Sam7]* | |
| Strain, strain background (*E. coli*) | Hfl-1 | PMID: 4352176 | *hfl-1, fhuA2::IS2, lacY1, tsx-1, glnX44, gal-6, xyl-7, mtlA2, mut-14* | |
| Strain, strain background (*E. coli*) | LE392 | PMID: 6291786 | *hsdR514 (rk–, mk+), glnX (supE44), tyrT (supF58), Δ(codB-lacI)3, galK2, galT22, metB1, trpR55* | |
| Strain, strain background (*E. coli*) | BW14879 | PMID: 2160940 | *pMW11 Muc62 Δ(lac)X74, Δ(phoA532 PvuII) phn(EcoB), arcA1655, fnr-1655* | B. Wanner, Purdue University |
| Strain, strain background (*E. coli*) | BW5104 | PMID: 2160940 | *Mu-1 Δlac169, creB510, hsdR514* | B. Wanner, Purdue University |
| Strain, strain background (*E. coli*) | RJ3960 | This work | BW5104 λR *mal* | |
| Strain, strain background (*E. coli*) | RJ3388 | Laboratory collection | BL21 (DE3) *endA::tet8, fis::str/spc-985* | |

*Continued on next page*

*Continued*

| Reagent type (species) or resource | Designation | Source or reference | Identifiers | Additional information |
|---|---|---|---|---|
| Strain, strain background (*E. coli*) | RJ3431 | Laboratory collection | BL21 (DE3) *metC*::Tn*10* | |
| Recombinant DNA reagent | See *supplementary file 2* | | | |
| Sequence-based reagent | See *supplementary file 3* | | | |
| Peptide, recombinant protein | DNase I | Thermo Fisher, Waltham, MA | Catalog number: EN0521 | |
| Peptide, recombinant protein | Exonuclease III | NEB, Ipswich, MA | Catalog number: M0206L | |
| Peptide, recombinant protein | Proteinase K | Roche, Germany | Catalog number: 03115828001 | |
| Peptide, recombinant protein | Trypsin | Promega, Madison, WI | Catalog number: V511A | |
| Commercial assay or kit | Sequenase Quick-Denature Plasmid Sequencing Kit | Affymetrix, Santa Clara, CA | Catalog number: 70140 | |
| Commercial assay or kit | Coomassie Protein Assay Reagent | Thermo Fisher, Waltham, MA | Catalog number: 1856209 | |
| Chemical compound, drug | Dimethyl sulfate | Thermo Fisher, Waltham, MA | Catalog number: AC430831000 | |
| Chemical compound, drug | Piperidine | Sigma-Aldrich | Catalog number: 10409–4 | |
| Chemical compound, drug | Diamide | Sigma-Aldrich | Catalog number: 87751 | |
| Chemical compound, drug | AEBSF | Gold Biotechnology | Catalog number: A-540–1 | |
| Software, algorithm | ImageQuant | GE Healthcare | RRID:SCR_014246 | |
| Software, algorithm | PyMOL Molecular Graphics System | Schrodinger, LLC | RRID:SCR_000305 | https://pymol.org/2/ |
| Software, algorithm | Protein Prospector /MS-Digest | http://prospector.ucsf.edu/prospector/cgi-bin/msform.cgi?form=msdigest | RRID:SCR_014558 | |
| Software, algorithm | Phyre2 | PMID: 25950237 | RRID:SCR_010270 | www.sbg.bio.ic.ac.uk/phyre2/ |
| Software, algorithm | XDS | PMID: 20124692 | RRID:SCR_015652 | http://xds.mpimf-heidelberg.mpg.de/ |
| Software, algorithm | PHASER | PMID: 19461840 | RRID:SCR_014219 | |
| Software, algorithm | SHELX | doi.org/10.1107/S0021889804018047 | RRID:SCR_014220 | |
| Software, algorithm | Coot | PMID: 15572765 | RRID:SCR_014222 | https://www2.mrc-lmb.cam.ac.uk/personal/pemsley/coot/ |

*Continued on next page*

*Continued*

| Reagent type (species) or resource | Designation | Source or reference | Identifiers | Additional information |
|---|---|---|---|---|
| Software, algorithm | Phenix | PMID: 20124702 | RRID:SCR_014224 | https://www.phenix-online.org/ |
| Software, algorithm | Buster | PMID: 22505257 | RRID:SCR_015653 | https://www.globalphasing.com/buster/ |
| Software, algorithm | CCP4 | PMID: 21460441 | RRID:SCR_007255 | http://www.ccp4.ac.uk/ |
| Software, algorithm | Procheck | doi.org/10.1107/S0021889892009944 | RRID:SCR_006511 | https://www.ebi.ac.uk/thornton-srv/software/PROCHECK/ |
| Software, algorithm | Clustal Omega | https://www.ebi.ac.uk/Tools/msa/clustalo/ | RRID:SCR_001591 | |

## Strains and plasmids

*E. coli* strain genotypes are given in *Supplementary file 1*. ISC*1926* was amplified from *S. islandicus pyrE::ISC1926* genomic DNA (gift of D. Grogan), and IS*1535* was amplified from *Mycobacterium tuberculosis* H37Rv genomic DNA (gift of D. Eisenberg). Synthetically-derived IS*607 orfA* and *orfB* sequences (*E. coli* codon optimized, Genewiz, South Plainfield, NJ), along with all plasmids used in this work and details of their constructions, are given in *Supplementary file 2* and *3*.

## Transposition assays

RJ1224 (*recA λbbnin* [*λcI857 b515 b519 nin5 S$_{am}$7*]) containing pBR322 with an IS*607-tet* derivative was grown at 30°C in 2 x YT and 10 µg/ml tetracycline. λ lysates were obtained upon shifting the culture to 42°C for 20 min and then to 37°C for 3 hr to allow phage development. Lysates were titered on LE392 (*supF*) and used to transduce early stationary phase LB cultures of the high frequency lysogenizing strain Hfl-1 (*Belfort and Wulff, 1973*) at a multiplicity of infection of about 0.3. After 20 min at 30°C, 2 volumes of LB were added, and incubation continued for 60 min. Cells were plated onto LB +10 µg/ml tetracycline, and the number of Tet$^R$ (Amp$^S$, temperature-sensitive) transductants per plaque forming unit (PFU) were scored as transposition events.

λ genomic fragments containing IS*607-tet* were transferred to plasmids for DNA sequencing by the in vivo mini-Mu cloning method (*Groisman and Casadaban, 1986*). Lysates of the Hfl-1 λ*bbnin*:: IS*607-tet* transductants were used to lysogenize BW14879 containing the mini-Mu cloning plasmid pMW11 (str/spc$^R$) and Muc62 (*Metcalf et al., 1990*). Mini-Mu lysates were prepared by thermal-induction and used to infect RJ3960, a λ$^R$ derivative of BW5104 selected as a maltose non-fermenting survivor after λcI$^-$ *b221* infection. After growth for 60 min at 30°C the cells were plated on LB +10 µg/ml tetracycline and 25 µg/ml streptomycin. Plasmid DNA from Tet$^R$, Str$^R$ colonies were sized on agarose gels, and plasmids < 15 kb were subjected to DNA sequencing using primer oRJ878 that reads out from the left end of IS*607*. The sequence identified the insertion position on the λ genome, and insertion-specific λ primers flanking the transposon were then used to amplify the region from the original λ::IS*607-tet* lysate as a template and to sequence the right junction using primer oRJ879 that reads out from the right end of IS*607-tet*. All amplicon sizes were consistent with simple insertions.

## Purification of TnpA and TnpA-CTD

TnpA proteins were expressed in RJ3388 in 2xYT at OD$_{600}$ = 1 with 0.4 mM IPTG for ~16 hr at 15°C. Cells expressing full-length proteins were lysed in 25 mM MES-NaOH, pH 6.0, 300 mM NaCl, 5 mM β-mercaptoethanol (βME), 5 mM EDTA, and 10% glycerol by three passes through a French Press. Clarified extracts were batch incubated with SP Sepharose Fast Flow resin (GE Healthcare, Chicago, Illinois) for 2 hr at 4°C, the resin was washed extensively with lysis buffer containing 400 mM NaCl,

and protein was eluted with 50 mM HEPES, pH 7.5, 1 M NaCl, 10% glycerol, and 5 mM βME. The partially purified TnpA was then bound to Ni-NTA agarose (Goldbio, St. Louis, Missouri) in the same buffer. The resin was washed with Buffer A (25 mM HEPES, pH 7.5, 1 M NaCl, 5 mM βME, and 10% glycerol) + 50 mM imidazole, and TnpA was eluted with Buffer A + 500 mM imidazole. Batch chromatography was used to avoid protein precipitation upon elution. TnpA was dialyzed into storage buffer (25 mM HEPES, pH 7.5, 1 M Na acetate, 5 mM βME, and 50% glycerol) and stored at −20°C or at −80°C after quick freezing.

RJ3388 expressing the TnpA-CTD were lysed by French Press in Buffer A (50 mM MOPS, pH 7.0, 1 M NaCl, 25 mM imidazole, 5 mM βME, and 10% glycerol). For Se-methionine (Se-met) labeling, RJ3431 (metC) containing pRJ3347 was grown in M9 glucose +20 µg/ml methionine to an $OD_{600}$ = 1.5. Cells were chilled and transferred to M9 glucose, incubated for 20 min at 15°C followed by addition of 60 µg/ml Se-met and 0.4 mM IPTG. Clarified extracts were incubated with Ni-NTA resin, washed in Buffer A + 50 mM imidazole, and eluted in Buffer B (50 mM MES-NaOH, pH 6.0, 5 mM βME, 10% glycerol) plus 500 mM NaCl and 250 mM imidazole. Eluted proteins were mixed with an equal volume of Buffer B and then incubated with SP Sepharose Fast Flow for 2 hr. The resin was washed with Buffer B + 400 mM NaCl, and near pure TnpA-CTD eluted in Buffer B + 1 M NaCl. The protein was concentrated with an Amicon Ultra-15 centrifugal filter (3 K Da cutoff, MilliporeSigma, Burlington, MA) and applied to a Superdex 75 (16/600; GE Healthcare) column on an FPLC in Buffer B + 1 M NaCl. The peak fractions containing the CTD were pooled, exchanged into crystallization buffer, and concentrated.

## Domain mapping by limited proteolysis and mass spectrometry

IS607 and IS1535 TnpA (3 µg) were incubated in 20 µl 25 mM HEPES (pH 7.5), 300 mM NaCl, 10% glycerol and 5 mM 2-mercaptoethanol at 37°C with 50 ng trypsin (Promega, Madison, WI) for varying times up to 30 min. TnpA$^{ISC1926}$ reactions were identical except that 10 mM $CaCl_2$ was included in the cleavage buffer, and 100 ng of trypsin was added. Proteolysis reactions were quenched with 5 mM AEBSF (4-(2-aminoethyl)benzenesulfonyl fluoride hydrochloride; Sigma-Aldrich, St. Louis, MO), and subjected to 18% SDS-PAGE in Tricine buffer with 10% glycerol in the separating gel and stained with Coomassie Blue. Aliquots were analyzed by MALDI-TOF-MS on an Applied Biosystems Voyager DE-STR instrument operated in positive ion mode with and without the reflectron. Upon testing several matrices, sinapinic acid was found to yield the best mass spectra. Peptide molecular weights were compared to all trypsin cleavage products calculated for the protein using the MS-Digest tool in Protein Prospector (http://prospector.ucsf.edu/prospector/cgi-bin/msform.cgi?form=msdigest) to determine most likely endpoints.

## Electrophoretic (gel) mobility shift assays

TnpA$^{IS1535}$ binding reactions were performed in 20 µl 25 mM HEPES, pH 7.5, 150 mM Na acetate, 5 mM Mg acetate, 1 mM DTT, 500 µg/ml BSA, 5% glycerol, 25 µg/ml sonicated salmon sperm DNA (Rockland, Limerick, PA)+1 nM $^{32}$P-labeled DNA probe. DNA probes were generated by PCR with LE or RE specific primers using pRJ3234 (IS1535 LE) or pRJ3348 (IS1535 RE) as the template (Supplementary file 2 and 3) and PAGE purified. The standard 149 bp LE probe (91 bp LE side, 58 bp host side) used in Figures 3 and 7 was generated using oRJ839 and oRJ840. A portion was end-labeled with γ-$^{32}$P-ATP (Perkin Elmer, Waltham, MA) and polynucleotide kinase (NEB) and free label removed with a G-50 Micro column (GE Healthcare). Labeled probe was added to unlabeled probe to generate 1 nM in the binding reaction. Freshly diluted TnpA in 25 mM HEPES, pH 7.5, 1 M Na acetate, 1 mM DTT, 500 ug/ml BSA, and 20% glycerol was added to the binding mixture and typically incubated at 37°C for 60 min before applying to a 6% polyacrylamide (acrylamide:bisacrylamide 37.5:1) in 25 mM Tris-acetate, pH 7.5, and 1 mM Mg acetate (gel and running buffer). Electrophoresis was typically at 3.5 v/cm for 12 hr at 23°C. TnpA$^{IS1535}$ proteins were oxidized for disulfide cross-linking by incubation at 4°C overnight in 25 mM HEPES, pH 7.5, 1 M Na acetate, 20% glycerol and 0.2 mM diamide, and the binding buffer contained 0.2 mM diamide in place of DTT. A Typhoon phosphorimager was used for image acquisition, and analysis was performed with ImageQuant (GE Healthcare).

## Nuclease and chemical probing of TnpA complexes

Binding reactions were the same as for the EMSAs except that the labeled probe was generated by amplifying pRJ3234 (IS*1535* LE), pRJ3348 (IS*1535* RE) (*Figure 4* and *Figure 8—figure supplement 1*) or pRJ3352 (*Figure 6*) with 5'-labeled oRJ880 or oRJ881. After 60 min incubation at 37°C with TnpA$^{IS1535}$, DNase I (0.02 u, Thermo Fisher, Waltham, MA) or Exonuclease III (10 u, NEB, Ipswich, MA) was added for 30 s or 5 min, respectively. Reactions were quenched with 150 mM Tris-HCl, pH 8.5, 10 mM CDTA, 0.8% SDS, and 12.5 µg/ml proteinase K and incubated 10 min at 65°C. The DNA was ethanol-precipitated, dissolve in formamide-NaOH dye and electrophoresed through 6% acrylamide-urea sequencing gels in TBE. Dimethyl sulfate reactions (10 mM, 30 s) under the same binding conditions and DNA cleavage with piperidine were performed essentially as described (*Shaw and Stewart, 1994*). Sequence ladders were generated using the Sequenase Quick-Denature Plasmid Sequencing Kit (Affymetrix, Santa Clara, CA).

## Crystallization and structure determination

The best diffracting crystals of TnpA$^{ISC1926}$ CTD were obtained using the hanging drop method by mixing equal volumes of a 10 mg/ml protein solution in 20 mM MOPS, pH 7.0, 100 mM Na-acetate, 0.1 mM DTT with a reservoir solution containing 8% (v/v) tacsimate, pH 4.0, and 20% (w/v) PEG3350. Crystals grew at 25°C, and although additional cryoprotectants were screened, they show no increase in diffraction relative to the drop solution alone. For TnpA$^{IS1535}$ CTD, optimal crystals were grown by mixing equal volumes of a 5 – 9 mg/ml protein solution in 0.3 M sodium acetate, pH 5.0, and 1 mM TCEP with a reservoir solution containing 0.2 M sodium citrate + 20% (w/v) PEG3350. Crystals were cryoprotected in reservoir solution plus 30% glycerol.

All X-ray diffraction data were collected at 100 K at the Advanced Photon Source (Chicago IL) beamline 24-ID-C on a DECTRIS-PILATUS 6M detector. TnpA$^{ISC1926}$ CTD data were collected to 2.9 Å and integrated and scaled with XDS (*Kabsch, 2010*). The phases were solved by molecular replacement with PHASER (*McCoy et al., 2007*) using 3LHK chain D as the search model. Model building and refinement were performed using Coot (*Emsley and Cowtan, 2004*), PHENIX (*Adams et al., 2002*), and BUSTER (*Smart et al., 2012*). TnpA$^{IS1535}$ CTD native and Se-met data were both collected to 2.5 Å resolution, and integrated and scaled using XDS. MAD phases were calculated from six selenium atoms with HKL2MAP (*Pape and Schneider, 2004*). Automatic model building was performed with BUCCANEER (*Winn et al., 2011*), which traced approximately 90% of the two chains. This model was then used to continue model building and refinement on the native dataset using Coot and BUSTER. X-ray data and refinement statistics are given in *Table 1*; the PDB code for the TnpA$^{ISC1926}$ CTD is 6DGC and for TnpA$^{IS1535}$ CTD is 6DGC. Molecular graphics images of the structures were produced with PyMOL (Schrödinger, https://pymol.org/2/).

## Modeling

Structure models of the N-terminal domains were generated by Phyre2 (*Kelley et al., 2015*). A structural model of an intact TnpA$^{ISC1926}$ dimer was generated from the Phyre2 model of the TnpA$^{ISC1926}$ NTD (residues 12 – 61, *Figure 2—figure supplement 2C*) linked to residue 65 of the CTD by the native residues Arg-Glu-Glu using Coot. The NTD was docked onto a DNA model (3DNA, [*Lu and Olson, 2003*]) of the IS*1535* LE sequence with the aid of the closely related RacA-DNA complex (*Figure 2—figure supplement 2C*) and DMS protection data (*Figure 8—figure supplement 1*) using PyMOL and Coot.

## Acknowledgement

We are grateful to Dennis Grogan (University of Cincinnati) for providing us with genomic DNA of *S. islandicus pyrE::ISC1926* and David Eisenberg (UCLA) for genomic DNA of *M. tuberculosis*. We thank Rachel Ogorzalek Loo (UCLA), and Nuraly Avliyakulov and Michael Haykinson (UCLA), for performing mass spectrometry. The UCLA X-ray core facility is supported in part by the Department of Energy grant DE-FC0302ER63421. We thank the Northeastern Collaborative Access Team beamline NECAT ID-24 at the Advanced Photon Source of Argonne National Laboratory, which is supported by National Institutes of Health grants P41 RR015301, S10 RR029205, and P41 GM103403. Use of

the Advanced Photon Source is supported by the Department of Energy under Contract DE-AC02-06CH11357. This work was supported by NIH grant GM038509 to RCJ.

## Additional information

### Funding

| Funder | Grant reference number | Author |
|---|---|---|
| National Institute of General Medical Sciences | GM038509 | Wenyang Chen<br>Sridhar Mandali<br>Stephen P Hancock<br>Pramod Kumar<br>Reid C Johnson |

The funders had no role in study design, data collection and interpretation, or the decision to submit the work for publication.

### Author contributions

Wenyang Chen, Conceptualization, Investigation, Methodology, Writing—review and editing; Sridhar Mandali, Conceptualization, Investigation, Methodology; Stephen P Hancock, Investigation, Visualization, Methodology, Writing—review and editing; Pramod Kumar, Michael Collazo, Investigation; Duilio Cascio, Resources, Supervision, Investigation, Methodology, Writing—review and editing; Reid C Johnson, Conceptualization, Resources, Supervision, Funding acquisition, Investigation, Visualization, Methodology, Writing—original draft, Project administration

### Author ORCIDs

Wenyang Chen (iD) http://orcid.org/0000-0003-3035-1496
Stephen P Hancock (iD) https://orcid.org/0000-0003-4205-7913
Reid C Johnson (iD) http://orcid.org/0000-0002-5562-1934

### Decision letter and Author response

Decision letter https://doi.org/10.7554/eLife.39611.031
Author response https://doi.org/10.7554/eLife.39611.032

## Additional files

### Supplementary files

• Supplementary file 1. *E.coli* strains used in this work.
DOI: https://doi.org/10.7554/eLife.39611.022

• Supplementary file 2. Plasmids used and constructed in this work.
DOI: https://doi.org/10.7554/eLife.39611.023

• Supplementary file 3. Oligonucleotides used in this work.
DOI: https://doi.org/10.7554/eLife.39611.024

• Transparent reporting form
DOI: https://doi.org/10.7554/eLife.39611.025

### Data availability

Diffraction data have been deposited in the PDB under accession codes 6DGB and 6DGC.

The following datasets were generated:

| Author(s) | Year | Dataset title | Dataset URL | Database, license, and accessibility information |
|---|---|---|---|---|
| Hancock SP, Kumar P, Cascio D, Johnson RC | 2018 | Crystal structure of the C-terminal catalytic domain of ISC1926 TnpA, an IS607-like serine recombinase | www.rcsb.org/structure/6DGC | Publicly available at the RCSB Protein Data Bank (accession no: 6DGC) |

Hancock SP, Chen WY, Cascio D, Johnson RC | 2018 | Crystal structure of the C-terminal catalytic domain of IS1535 TnpA, an IS607-like serine recombinase | www.rcsb.org/structure/6DGB | Publicly available at the RCSB Protein Data Bank (accession no: 6DGB)

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
