## [Decision Letter]

Thank you for submitting your article "Multiple serine transposase dimers assemble the transposon-end synaptic complex during IS*607*-family transposition" for consideration by *eLife*. Your article has been reviewed by Gisela Storz as the Senior Editor, a Reviewing Editor, and three reviewers. The reviewers have opted to remain anonymous.

The reviewers have discussed the reviews with one another and the Reviewing Editor has drafted this decision to help you prepare a revised submission.

This is an important and well-executed paper on IS607 family transposition. Both the experimental work and its description are well-done, clear, and of a high standard. The manuscript represents an important contribution to the field.

Overall, the quality and thoroughness of the science in this paper is first rate. Some revision based on the comments of the reviewers will strengthen the manuscript in how the results are interpreted and how the models are to be favored or disfavored. The work sets the stage for further testing of models and examining mechanisms for IS607 family transposition. Although the list of comments is extensive, that authors should note that none of the reviewers have insisted on any further experiments by the authors; if they can supply any more experimental information that they may have on the points made by the reviewers (e.g., the activity of LE-LE complexes), then addition of such data would improve the paper, but is not essential.

Otherwise, despite the longer than normal list of comments below, the authors should be aware that most of the questions arose from an interest in their work, rather than from skepticism of their conclusions. In the end, all of the changes required are simply additions or clarifications of the text (e.g., the Discussion section). Consequently, it should be relatively easy for the authors to modify the text to accommodate these suggestions and improve the manuscript within two months.

[Please note that it was difficult to summarize the reviewers' comments without including most of the full review to provide the proper context of their comments. Consequently, what follows is an atypical *eLife* review of a manuscript that is in principle acceptable for publication and, instead of an editorial summary, contains the verbatim comments of the reviewers. The discussion amongst the reviewer is a summary of the major points that should be clarified in the revised manuscript, although the authors should consider the full scope of the reviewers’ comments.]

Summary of reviewers' discussion and the specific comments to be addressed (please see the full set of comments by the reviewers for context):

1) Both reviewers 1 and 3 were in complete accord with principal comments 1-3 as expressed by reviewer 1 (see related comments by reviewer 3).

2) Both reviewers 1 and 3 agree that the foot-printing data is extensive and thorough, and characterizes the interactions of TnpA dimers with individual binding elements within each transposon end, and the allosteric/cooperative contributions to binding. One concern is whether this binding characterization is sufficient to draw strong conclusions regarding paired end complex (PEC) formation (or synapsis of the ends), which is the principal question that the authors set out to solve.

3) Both reviewers 1 and 3 found the model for cooperative TnpA assembly at the left end followed by capture of the unbound right end somewhat unsubstantiated by the experimental data. Of course, this is a model at this stage. Both reviewers 1 and 3 were not sure that the present results rule out PEC formation by TnpA-bound left and right ends, with the pairing stabilizing the TnpA dimers bound to the right end. This should be addressed or the conclusions qualified.

4) Both reviewers 1 and 3 thought that the authors may want to restate their position on why they consider rotation unlikely rather than rule out this possibility. Although they agree with the authors that subunit rotation is unlikely, they nonetheless, agree that subunit rotation cannot be dismissed as a mechanism based on the data presented. Perhaps the authors could expand this discussion a bit by explaining their reasoning and noting that their data do not rule it out. In this context, the authors should consider: (1) The dimerization interface does not involve helix-E and therefore subunit rotation would have to occur via a different interface compared to superfamily members with C-terminal DBDs. (2) I don't see why anything needs to rotate when the transposon is excised from the host.

5) Both reviewers 2 and 3 agree that the comments on Figure 8 by reviewer 2 models based on affinity differences in binding.

6) Both reviewers 2 and 3 agree thought that the role (or lack thereof) for helix E in PEC formation is worth consideration from the structure perspective.

7) The question of the functional competence of identical sequences at left and right ends of the transposon placed in the proper orientation is a relevant question that all three reviewers raise.

8) Finally, an additional topic that the authors may want to briefly discuss at the end is how their model might accommodate binding to target DNA. Presumably, another catalytic dimer equivalent needs to be provided. This would be speculative, but would relate the results of the paper back to the overall transposition pathway for readers.

*Reviewer #1:*

Chen et al. report characterization of the transposase-DNA interactions involved in transposition by members of the IS607 family of transposons, along with two new structures of transposase catalytic domains.

This is an excellent manuscript which provides substantial and much-needed information about how IS607-family serine transposases interact with their transposon ends to promote strand transfer. The work is presented very clearly and the manuscript is very well written throughout. Also, some difficult experimental work has been carried out to a high standard – in particular, the footprinting data are very nice. I think that the manuscript will be an important contribution to the field that will be very useful to others who wish to investigate these unusual transposition systems further. I have some comments on specific issues which I hope the authors will address (see below), but these are all of a relatively minor nature.

1) Much of the biochemical analysis is done on an IS1535 paired-ends complex (PEC) comprising two left ends (LE) bound together by the transposase. However, one would expect that for natural transposition, the active complex would be LE-RE, which is apparently less stable in the authors' assays. Is there any evidence that the LE-LE PEC is an active intermediate? For example, can transposition of an artificial element with two LEs be observed?

2) Results section and Discussion section. The authors propose that a single-end complex captures an unbound end to generate a PEC. Could the authors make their justification of this hypothesis clearer? Can they exclude both ends being bound by transposase? If their proposal is correct, formation of the PEC should be inhibited by higher concentrations of TnpA.

3) Last paragraph of Discussion section. The authors propose that excision and integration by this family of transposases might not involve a 'subunit rotation' mechanism as is proposed for related serine site-specific recombinases – which may be true, but I don't think that any of the data presented here argue strongly against a rotation mechanism.

*Reviewer #2:*

This is an interesting paper that represents an advance in understanding how IS607 family transposition occurs. The experimental support for formation of a TnpA nucleoprotein filament on LE DNA and a much smaller complex on RE DNA are strong. The data showing highly cooperative PEC formation between two LEs is also convincing.

The models in Figure 8 suggest that the IS607 serine recombinases acting at 'accessory sites' do so in a manner that is quite different from what the catalytic dimers must be doing. This isn't the case for the other small SR dimers, which all bind DNA in roughly similar ways, regardless of whether or not they are catalytic. Since the catalytic dimers of TnpA likely function as well-positioned, but non-specific nucleases in the excision reaction, perhaps this explains why the active sites are kept away from DNA on the filament (according to the model proposed).

One implication of the Figure 8 models is that 8A must not be much higher affinity vs 8B. If this weren't the case, I don't see how a LE complex could ever form; the intramolecular wHTH-DNA interactions would dominate. It seems like this should be a testable prediction.

I was surprised that the E-helices are not required for PEC formation; this was an important result that limits the types of interactions likely to occur on the filament vs between LE and RE.

With a strong focus on complexes formed on and between LE DNA, I was surprised that the activity of LE alone was not mentioned. Is a LE-TnpA-tet-LE(inv) construct active or toxic in vivo? Is there any sign of cleavage or covalent intermediate in the PECs formed at high TnpA?

*Reviewer #3:*

This paper by Chen et al., is a detailed structural and biochemical analyses of the DNA-protein interactions involved in the assembly of the high-order structures that mediate IS607 family transposition. This family of transposons deviates from the classical definition of transposons in that they do not harbour the canonical 'inverted repeat sequences' at their termini, and do not generate target site duplications at their insertion points. Transposition requires a single protein dimer (the transposase), coded for the transposase, that forms functional oligomers on cognate DNA elements. The catalytic pocket of the transposase is typical of well characterized serine site-specific recombinases, which carry out the strand cleavage and strand joining steps of recombination by transesterification chemistry. By contrast, the classical transposases perform strand cleavage by hydrolysis, and strand transfer (joining) by transesterification. The consensus dinucleotide (GG) at the insertion site in the recipient DNA (also the left and right junctions of the insertion) is consistent with the typical serine recombination mechanism in which the exchanged region is 2 bp long.

The principal conclusion from this study is that oligomeric assembly of the transposase at one end of the transposon containing multiple binding elements captures the other end containing fewer such elements to form a paired end complex (PEC), or a 'transpososome'. PEC formation does not require a change in the dimeric structure of the C-terminal catalytic domain, even though some remodeling of the E-helix within this domain seems to be necessary. The chemical activity of the transposase would involve some significant movement within the observed dimer structure (without DNA) in which the catalytic serine residues are too far apart to attack their target phosphodiester bonds.

This is a well written paper with an extensive set of data presented clearly in figures and text, and interpreted carefully. The structural view of the transposition complex emerging from the present studies will certainly provide the foundation for further testing mechanistic models for the chemical steps of the reaction carried out by this complex. The following comments may be considered if the authors wish to view their system from the broader perspective of phosphoryl (nucleotidyl) transfer by transesterification chemistry rather than from the narrower perspective of transposition.

1) The lack of target DNA duplication in the IS607 family is interesting from a purely transposition point of view, and based on precedent. The length of the duplication reflects the spacing of the nucleophiles (3'-hydroxyls in canonical transposition) and in turn the staggered insertion of the transposon ends at the target site. Depending on the extent of replication/repair, the nucleotides spanning the gap generated by the insertion, or these nucleotides plus the entire transposon, can be duplicated to give either a simple insertion or a co-integrate. Since the length of the duplication varies among different families, a few bp to as many as ~30 bp for CRISPRs, why not think of the IS607 family as a '0 bp' duplication family?

2) In some sense, the differences between conservative site-specific recombination, DNA transposition and even homologous recombination are often less marked than they are purported to be, if considered from a purely mechanistic standpoint. Given the rather limited chemical repertoire available for biological catalysis, how many different ways can one break or form a phosphodiester bond in DNA or RNA-rather few. Topoisomerases, conservative site-specific recombinases, enzymes that carry out homologous recombination and DNA/RNA polymerases illustrate he limited number of ways that this chemistry can be performed under different biological contexts. Juggling these mechanisms to reach similar genetic outcomes, or using the same mechanism to bring about distinct genetic rearrangements, is expected to be the rule rather than the exception. The authors may wish to tone down their characterization of IS607 family as an outlier among transposons.

3) Given that the transposase uses a serine nucleophile-based mechanism, the strand cleavage and the strand transfer steps of transposition must follow transesterification chemistry. Hence the reaction is more akin to site-specific recombination. One could think of the insertion of the transposon as recombination-mediated cassette exchange (RMCE), although a non-standard one. The donor cassette (transposon) is flanked by two recombination sites as in normal RMCE, but the target site has only one equivalent site, so the excised cassette in exchange for transposon insertion is a '0 bp cassette' (on par with 0 bp target duplication during transposition).

I believe that the 'cut-and-paste' mechanism that authors propose in the 'Discussion' is more or less the same as the non-reciprocal RMCE. An implication of this mechanism is that the donor DNA is likely to be repaired efficiently without addition or loss of DNA. Is this true from in vivo assays? It is probably known and mentioned here, and I might have somehow missed it.

Specific Comments:

1) Summary: Is the last sentence 'We posit-PEC recruits a chemically-active conformer of TnpA -.' justified by the data? Is it not possible that the TnpA dimer (tetramer?) adjacent to the scissile phopshates acquires cleavage competence within the assembled PEC? The sentence, as it stands now, suggests that there are inactive and active TnpA dimers in solution and the assembled PC selectively enlists active dimers to the cleavage site.

2) Figure 2 and Figure 3 (figure supplements) and text on structural data. The domain characterization of TnpA, the structural determination (C-terminal domain) and the relevant comparisons to serine recombinase catalytic domains are nicely organized and quite helpful. The wide spacing between the catalytic serine residues within the dimer unbound to DNA is not surprising, given prior examples.

In the Discussion section, there is a suggestion that the cleavage might occur in trans. Does this refer to cleavage within the transposon ends or cleavage of a transposon end and the target? Will the serine arrangement in the dimer seen in the structure be consistent with the latter mode of cleavage?

3) Figure 4 to Figure 8 (and figure supplements) and corresponding text. This section represents the heart of the paper, with extensive characterization of TnpA binding to the repeat elements of the left and right ends by EMSA, DNase I and exonuclease foot printing etc. The conclusion from the cumulative assays is the formation of the nucleation of a TnpA filament at the left end followed by capture of the right end to form the transposition synapse (paired end complex; PEC).

Figure 3 and supplements show the ability of left end to form PEC, as well as theinability of the right end to do so. The assay uses two separate DNA fragments, one as the radio-labeled probe for binding the second as an unlabeled fragment for capture into PEC. If two right ends are present on the same DNA fragment, do the ends come together in a PEC?

Are all the binding elements a-d present at the left end required for PEC formation with right end? If the left end is replaced by a copy of the right end, will transposition occur?

Will additional binding elements at the right end be inhibitory to PEC formation and transposition? Here, one would expect filament formation at both ends.

4) Continued from 3. According to the PEC model, the ends are bridged by a series of TnpA dimers initially lined up at the left end. Do the present results exclude the possibility that the PEC formation is mediated via dimers of TnpA dimers, bound at the left and the right ends?

Is it possible to think of the extra binding elements abutting those next to the scissile phosphates as accessory sites, by analogy to Tn3 resolvase? In such a scenario, the TnpA dimers bound to the accessory sites may allosterically activate the TnpA dimers that perform DNA cleavage/transfer. The positioning of the TnpA dimers within one of the two ends cannot possibly form a high-order topological arrangement. However, a functionally relevant inter-wrapping of DNA between the bound ends would seem plausible, and is rather fleetingly alluded to by the authors.

5) Target capture? While the authors have presented detailed arguments to highlight the importance of the PEC in initiating the chemistry of transposition, the potential means of target capture and the mechanism of cleavage at the insertion site is rather unclear. In the PEC-target complex, is there a TnpA dimer at the target site, or is target cleavage accomplished by TnpA associated with the PEC? Obviously, the experiments presented here do not address this question directly. However, it would be helpful to think about target capture/cleavage in light of existing transposition models.

Perhaps an additional figure expanding on the mechanistic implications of the binding model shown in Figure 8 in cleavage and strand transfer may be appropriate (even if speculative). For example, are the cut-out of the transposon, insertion into the target, and healing of the donor concerted events? Or, does transposon excision from the donor precede its integration into the target?

The authors seem to rule out DNA rotation following cleavage as part of the strand transfer event. Is this inference consistent with interface of the TnpA dimer in the structure? Can one ignore the possibility of rotation between cleaved transposon ends and target ends?

6) In summary, this is an important study that reveals a mode of transpososome assembly in the IS607 family that differs from the more conventional mechanisms of assembly that we are currently aware of. Many of the implications of the model proposed will be tested by in vitro reactions, which the authors are eminently capable of successfully performing.

---

## [Author Response]

Reviewer 1's comments 1-3 are addressed below in the context of the Editor's summary comments 1, 3, and 4.

1) Both reviewers 1 and 3 were in complete accord with principal comments 1-3 as expressed by reviewer 1 (see related comments by reviewer 3).3) Both reviewers 1 and 3 found the model for cooperative TnpA assembly at the left end followed by capture of the unbound right end somewhat unsubstantiated by the experimental data. Of course, this is a model at this stage. Both reviewers 1 and 3 were not sure that the present results rule out PEC formation by TnpA-bound left and right ends, with the pairing stabilizing the TnpA dimers bound to the right end. This should be addressed or the conclusions qualified.4) Both reviewers 1 and 3 thought that the authors may want to restate their position on why they consider rotation unlikely rather than rule out this possibility. Although they agree with the authors that subunit rotation is unlikely, they nonetheless, agree that subunit rotation cannot be dismissed as a mechanism based on the data presented. Perhaps the authors could expand this discussion a bit by explaining their reasoning and noting that their data do not rule it out. In this context, the authors should consider: (1) The dimerization interface does not involve helix-E and therefore subunit rotation would have to occur via a different interface compared to superfamily members with C-terminal DBDs. (2) I don't see why anything needs to rotate when the transposon is excised from the host.

1) The most important comment, reiterated by the editor, concerns the chemical activity of the IS*1535* TnpA complexes. This is an area of considerable ongoing effort for us. We have tried quite hard to obtain evidence of transposition of various *M. tuberculosis* IS*1535* transposon derivatives in *E. coli*, including *LE-tnpA-tet-RE* and *LE-tnpA-tet-LE* constructs. Unfortunately, sequence analyses of the rare candidates have thus far failed to confirm any true transposition events. Also, despite considerable effort, we have not yet obtained evidence for chemical activity in vitro (comment 4 of reviewer 2). There could be a number of reasons for this. As elaborated in the Discussion, our current thinking is that catalytically-active TnpA in an alternate folded state must be recruited onto the PEC at the transposon-host junction. This step may require co-translational or chaperone-assisted folding onto the PEC. This would have to occur in the heterologous *E. coli* cells in our in vivo assays. It is also possible that a host factor required for IS*1535* transposition is missing in *E. coli* or that IS*1535 tnpA* has become catalytically inactive over time, even though it retains the normal constellation of residues around the active site that are believed to be important for SR chemistry. For example, IS*1535* TnpA it may not be able to fold into the chemically-active conformation.

2) Are the footprinting assays really probing the PEC? Comparisons of the PEC assembly reactions with increasing TnpA by EMSAs (Figure 3) and by DNase I protections or Exo III stops in the footprinting reactions (Figure 4) show a close correspondence. Nevertheless, we directly assayed complexes formed in the footprint reactions by EMSA (removed an aliquot for EMSA from the footprint binding reactions immediately before nuclease digestion) and provided an example in Figure 4 figure supplement 1 in the original version. Note that at the amount of TnpA where the DNA is near fully protected from DNase digestion over the core motifs (Figure 4A), almost all of the DNA is within a PEC by EMSA (Figure 4—figure supplement 1). We could provide a similar EMSA example for a footprint reaction on LE: v54-20v (Figure 6), but believe this would be redundant (a standard EMSA for this mutant is presented in Figure 5—figure supplement 1C). It is possible that footprint signals at low TnpA concentrations, particularly in the more sensitive Exo III assays, may arise from some single-end coated complexes that may not be stable to electrophoresis in the EMSAs.

3) The reviewers (reviewer 1 principal comment 2) would like us to better justify why we propose that a single-end complex captures an unbound end to generate a PEC. Specifically, reviewer 1 asks whether high TnpA over end DNA concentrations inhibit PEC formation, which would be a direct prediction of the model. This prediction is born out in Figure 3A, and we noted this on p. 8: "Appearance of complex 2 is accompanied by a [similar] decrease of PECs" and have added a sentence emphasizing this point in the Discussion section. To better illustrate this point we have substituted Figure 3B with a plot that quantifies the complete TnpA titration range, showing a decrease in the number of PEC at high TnpA concentrations with a corresponding increase in the amounts of complex 2 (proposed to be single-end bound complexes). An expanded view of the lower end of the titration range that highlights formation of the PEC and was given in the original Figure 3B is now an insert within the Figure 3B panel.

We note in the Results section and elaborate in the Discussion section, that there is no evidence for a kinetic intermediate in the formation of the PEC, although one could argue that two bound ends could rapidly associate into a PEC and preclude detection at low TnpA concentrations. We believe the structural arguments for a single-end complex capturing an unbound end elaborated in the Discussion section and illustrated in Figure 8 add additional support. These considerations, together with points discussed in our response to principal comments 5 and 6, lead us to believe that our proposed PEC assembly model whereby one TnpA-bound end captures a free end is the most parsimonious explanation for the current information, but a more structurally complex model of synapsis by collision of two bound sites cannot be ruled out.

2) Both reviewers 1 and 3 agree that the foot-printing data is extensive and thorough, and characterizes the interactions of TnpA dimers with individual binding elements within each transposon end, and the allosteric/cooperative contributions to binding. One concern is whether this binding characterization is sufficient to draw strong conclusions regarding paired end complex (PEC) formation (or synapsis of the ends), which is the principal question that the authors set out to solve.

The authors may want to restate their position on why they consider rotation unlikely rather than rule out this possibility. We emphasize that we do not "rule out" the possibility of subunit rotation. We write: "We suggest that the element may be excised from the donor site and then insert into a target site using serine chemistry without strand transfer coupled to subunit rotation." As the reviewers would no doubt agree, any discussion on the nature of the target capture and strand transfer steps is speculation at this point. We currently have no data regarding the structural nature of the chemically-active protein, the target capture step, and the chemistry of transposon end cleavage and strand transfer reactions. In the absence of any data-driven basis for discussion, we question the value of describing multiple hypothetical scenarios for these steps. Nevertheless, we have re-written the last paragraph to slightly expand the discussion on target capture and implications for strand exchange by a subunit rotation reaction.

5) Both reviewers 2 and 3 agree that the comments on Figure 8 by reviewer 2 models based on affinity differences in binding.

The Figure 8A complex must not be much higher affinity than the Figure 8B complex. We agree and emphasize throughout that complex 1 is only observed at high TnpA concentrations. We imagine that a TnpA dimer dynamically associates with DNA in solution in the configuration represented by Figure 8B, but this complex is not stable to gel electrophoresis. However, in the context of multiple helically-phase motifs, a single dimer as in Figure 8B recruits other like dimers to form the filament represented by Figure 8C, which, in the absence of free LE DNA, is complex 2 that is at least somewhat stable to gel electrophoresis. Complex 1, which we are proposing to be represented by Figure 8A, appears on gels at much higher TnpA concentrations than required for PEC formation, even though by computer this dimer conformation readily docks onto DNA, and one might predict would be relatively stable. Complex 1 does not appear to effectively compete with PEC formation on LEs by EMSAs or by Exo III (solution) assays. We believe that there are important kinetic features at play in the assembly of the PEC: the slow step may be the initial nucleation of the array of dimers on a single end. Once the array with its constellation of unsatisfied binding sites is formed, it rapidly associates with a naked end to form a stable PEC. We have added this to the Discussion section.

6) Both reviewers 2 and 3 agree thought that the role (or lack thereof) for helix E in PEC formation is worth consideration from the structure perspective.

Consider the role (or lack thereof) for helix E in PEC formation. We believe the Reviewers are thinking about a potential model by which synapsis of two pre-bound ends occurs by an association of (remodeled) helix E regions, which could be considered unlikely because of weak PEC formation by the helix E deletion mutant. We agree and have added this point at the end of the Results section and in the Discussion section. The experimental evidence disfavoring this possible mechanism for PEC assembly is in line with our favored model of a single bound end capturing a free end.

7) The question of the functional competence of identical sequences at left and right ends of the transposon placed in the proper orientation is a relevant question that all three reviewers raise.

What is the functional competence of identical sequences at left and right ends of the transposon? Although we have discussed making IS*607* derivatives with two left or right ends to test the functional consequence of two identical ends on transposition in vivo, it has not been done. This is a good point, and we will now make testing this a priority. We note, however, that most IS607-family elements have multiple motifs variably distributed over relatively large segments at each end (e.g., Figure 1B) and that it is possible that the poor right end in IS*1535* may be an anomaly, as hinted at in the Discussion section.

8) Finally, an additional topic that the authors may want to briefly discuss at the end is how their model might accommodate binding to target DNA. Presumably, another catalytic dimer equivalent needs to be provided. This would be speculative, but would relate the results of the paper back to the overall transposition pathway for readers.

Target DNA binding and strand transfer into target DNA. As discussed above in point 4, how target capture and strand transfer occur are important issues that, unfortunately, we currently have no information on. We have incorporated a brief discussion of these issues along with subunit rotation in the last paragraph of the Discussion section. The focus of this paper is, of course, on the earlier step of paired-end complex formation.

Reviewer #3:

1) The lack of target DNA duplication in the IS607 family is interesting from a purely transposition point of view, and based on precedent. The length of the duplication reflects the spacing of the nucleophiles (3'-hydroxyls in canonical transposition) and in turn the staggered insertion of the transposon ends at the target site. Depending on the extent of replication/repair, the nucleotides spanning the gap generated by the insertion, or these nucleotides plus the entire transposon, can be duplicated to give either a simple insertion or a co-integrate. Since the length of the duplication varies among different families, a few bp to as many as ~30 bp for CRISPRs, why not think of the IS607 family as a '0 bp' duplication family?

Why not think of the IS607 family as a '0 bp' duplication family? Although the functional consequence is indeed a "0 bp duplication," the serine chemistry mechanism that presumably generates double strand cuts and joints using two active sites (dimer) on each duplex DNA is different from conventional mechanisms. We wonder whether it is useful to lump the different catalytic strategies/families together.

2) In some sense, the differences between conservative site-specific recombination, DNA transposition and even homologous recombination are often less marked than they are purported to be, if considered from a purely mechanistic standpoint. Given the rather limited chemical repertoire available for biological catalysis, how many different ways can one break or form a phosphodiester bond in DNA or RNA-rather few. Topoisomerases, conservative site-specific recombinases, enzymes that carry out homologous recombination and DNA/RNA polymerases illustrate he limited number of ways that this chemistry can be performed under different biological contexts. Juggling these mechanisms to reach similar genetic outcomes, or using the same mechanism to bring about distinct genetic rearrangements, is expected to be the rule rather than the exception. The authors may wish to tone down their characterization of IS607 family as an outlier among transposons.

We agree that only a handful of catalytic mechanisms exist that become tailored to generate specific types of DNA rearrangements through their unique synaptic complex architecture – indeed, the SRs are a prime example. On the other hand, we believe it is useful to highlight the unusual features of IS*607*-family elements in comparison to other transposable elements.

3) Given that the transposase uses a serine nucleophile-based mechanism, the strand cleavage and the strand transfer steps of transposition must follow transesterification chemistry. Hence the reaction is more akin to site-specific recombination. One could think of the insertion of the transposon as recombination-mediated cassette exchange (RMCE), although a non-standard one. The donor cassette (transposon) is flanked by two recombination sites as in normal RMCE, but the target site has only one equivalent site, so the excised cassette in exchange for transposon insertion is a '0 bp cassette' (on par with 0 bp target duplication during transposition).

*I believe that the 'cut-and-paste' mechanism that authors propose in the 'Discussion' is more or less the same as the non-reciprocal RMCE. An implication of this mechanism is that the donor DNA is likely to be repaired efficiently without addition or loss of DNA. Is this true from* in vivo *assays? It is probably known and mentioned here, and I might have somehow missed it.*

We agree that recombination-mediated cassette exchange (RMCE) has features related to our current view of the IS*607*-family transposition pathway. All of the IS*607* transposition products that we analyzed and that were reported by Kersulyte et al. are precise. On the other hand, there appear to be occasional interesting issues with joints among *Sulfolobus* ISC*1926*-like elements which may inform on the target insertion step. This needs more investigation and is beyond the scope of the present report on PEC formation.

Specific Comments:1) Summary: Is the last sentence 'We posit-PEC recruits a chemically-active conformer of TnpA -.' justified by the data? Is it not possible that the TnpA dimer (tetramer?) adjacent to the scissile phopshates acquires cleavage competence within the assembled PEC? The sentence, as it stands now, suggests that there are inactive and active TnpA dimers in solution and the assembled PC selectively enlists active dimers to the cleavage site.

Our currently favored model as "posited” in the summary is that an assembled PEC recruits a chemically-active conformer of TnpA to be positioned over the cleavage site, but we have no hard data for this. As the reviewer suggests in comment 4, an alternative model that is also plausible is that the PEC allosterically activates TnpA proteins that may be weakly bound over the transposon-host junction. We have added this possibility in the relevant part of the Discussion section. Note that the footprint assays on the LE indicate TnpA binds over the junction very weakly: by DNase I protection binding is not always evident, and binding has not been detected by Exo III.

2) Figure 2 and Figure 3 (figure supplements) and text on structural data. The domain characterization of TnpA, the structural determination (C-terminal domain) and the relevant comparisons to serine recombinase catalytic domains are nicely organized and quite helpful. The wide spacing between the catalytic serine residues within the dimer unbound to DNA is not surprising, given prior examples.In the Discussion section, there is a suggestion that the cleavage might occur in trans. Does this refer to cleavage within the transposon ends or cleavage of a transposon end and the target? Will the serine arrangement in the dimer seen in the structure be consistent with the latter mode of cleavage?

The suggestion in the Discussion section (noting the Boocock and Rice reference) that cleavage might occur in trans has been re-written to specify cleavage of the ends and/or the target. However, because the catalytic protomers must be in a very different structure than the solution and PEC-associated dimer, we really cannot firmly speak to this issue. In the Figure 8 models that are based on the solution dimer structures, the serines are >50 Å from the DNA to which they are bound.

3) Figure 4 to Figure 8 (and figure supplements) and corresponding text. This section represents the heart of the paper, with extensive characterization of TnpA binding to the repeat elements of the left and right ends by EMSA, DNase I and exonuclease foot printing etc. The conclusion from the cumulative assays is the formation of the nucleation of a TnpA filament at the left end followed by capture of the right end to form the transposition synapse (paired end complex; PEC).Figure 3 and supplements show the ability of left end to form PEC, as well as theinability of the right end to do so. The assay uses two separate DNA fragments, one as the radio-labeled probe for binding the second as an unlabeled fragment for capture into PEC. If two right ends are present on the same DNA fragment, do the ends come together in a PEC?Are all the binding elements a-d present at the left end required for PEC formation with right end? If the left end is replaced by a copy of the right end, will transposition occur?Will additional binding elements at the right end be inhibitory to PEC formation and transposition? Here, one would expect filament formation at both ends.

To be clear, our in vitro assays are primarily analyzing the robust assembly of two left ends into a PEC. Two right ends will weakly synapse (Figure 3E) and a left end will weakly synapse with a right end (Figure 3F). We have not tested left end deletions for their ability to synapse with the right end, nor added motifs to the right end to improve its activity, and have not tested an IS*1535* RE-RE construct for transposition.

4) Continued from 3. According to the PEC model, the ends are bridged by a series of TnpA dimers initially lined up at the left end. Do the present results exclude the possibility that the PEC formation is mediated via dimers of TnpA dimers, bound at the left and the right ends?Is it possible to think of the extra binding elements abutting those next to the scissile phosphates as accessory sites, by analogy to Tn3 resolvase? In such a scenario, the TnpA dimers bound to the accessory sites may allosterically activate the TnpA dimers that perform DNA cleavage/transfer. The positioning of the TnpA dimers within one of the two ends cannot possibly form a high-order topological arrangement. However, a functionally relevant inter-wrapping of DNA between the bound ends would seem plausible, and is rather fleetingly alluded to by the authors.

See responses to principal comments 3 and 6, and comment 1 concerning the PEC assembly and allosteric activation models. We do mention that the PEC may be assembled into a more inter-wrapped structure than depicted in Figure 8E in the Discussion section, Figure 8E legend, and Figure 8—figure supplement 3 legend.

5) Target capture? While the authors have presented detailed arguments to highlight the importance of the PEC in initiating the chemistry of transposition, the potential means of target capture and the mechanism of cleavage at the insertion site is rather unclear. In the PEC-target complex, is there a TnpA dimer at the target site, or is target cleavage accomplished by TnpA associated with the PEC? Obviously, the experiments presented here do not address this question directly. However, it would be helpful to think about target capture/cleavage in light of existing transposition models.Perhaps an additional figure expanding on the mechanistic implications of the binding model shown in Figure 8 in cleavage and strand transfer may be appropriate (even if speculative). For example, are the cut-out of the transposon, insertion into the target, and healing of the donor concerted events? Or, does transposon excision from the donor precede its integration into the target?The authors seem to rule out DNA rotation following cleavage as part of the strand transfer event. Is this inference consistent with interface of the TnpA dimer in the structure? Can one ignore the possibility of rotation between cleaved transposon ends and target ends?

See responses to principal comments 4 and 8 concerning target capture and subunit rotation.